# What Do You See in Common?
# Learning Hierarchical Prototypes over Tree-of-Life to Discover Evolutionary Traits

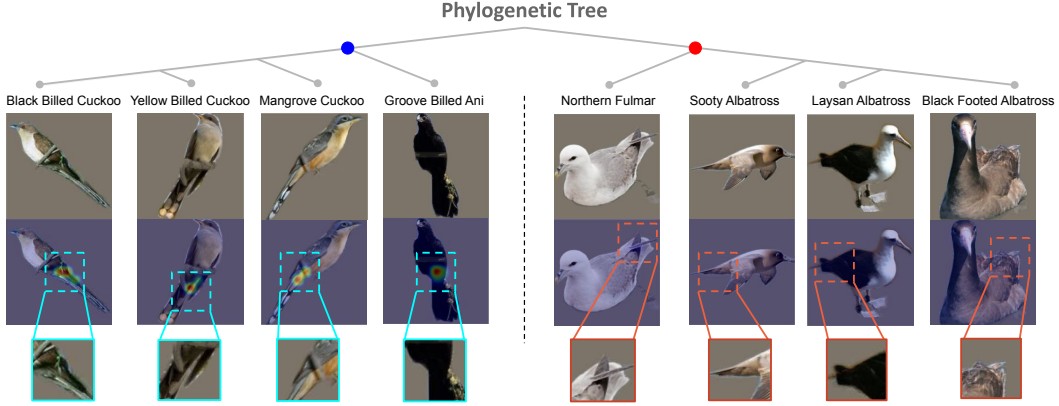

Figure 1: Sample images of bird species with zoomed-in views of learned prototypes along with their associated score maps. We consider the problem of finding evolutionary traits common to a group of species derived from the same ancestor (blue) that are absent in other species from a different ancestor (red). We can infer that descendants of the blue node share a common trait: *long tail*, absent from descendants of the red node.

## Abstract

A grand challenge in biology is to discover evolutionary traits—features of organisms common to a group of species with a shared ancestor in the tree of life (also referred to as phylogenetic tree). With the growing availability of image repositories in biology, there is a tremendous opportunity to discover evolutionary traits directly from images in the form of a hierarchy of prototypes. However, current prototype-based methods are mostly designed to operate over a flat structure of classes and face several challenges in discovering hierarchical prototypes, including the issue of learning over-specific features at internal nodes. To overcome these challenges, we introduce the framework of **H**ierarchy aligned **Com**monality through **P**rototypical **Net**works (**HComP-Net**). We empirically show that HComP-Net learns prototypes that are accurate, semantically consistent, and generalizable to unseen species in comparison to baselines on birds, butterflies, and fishes datasets.

## 1 Introduction

A central goal in biology is to discover the observable characteristics of organisms, or *traits* (e.g., beak color, stripe pattern, and fin curvature), that help in discriminating between species and understanding

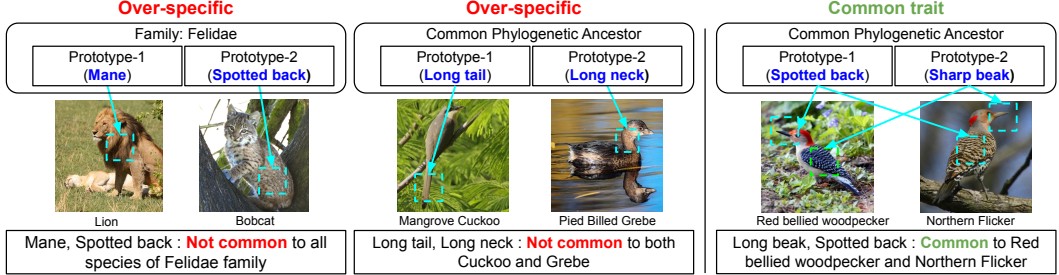

Figure 2: Examples to illustrate the problem of learning "over-specific" prototypes at internal nodes, which only cover one descendant species of the node instead of learning prototypes *common* to all descendants.

how organisms evolve and adapt to their environment [1]. For example, discovering traits inherited by a group of species that share a common ancestor on the tree of life (also referred to as the *phylogenetic tree*, see Figure 1) is of great interest to biologists to understand how organisms diversify and evolve [2]. The measurement of such traits with evolutionary signals, termed *evolutionary traits*, is not straightforward and often relies on subjective and labor-intensive human expertise and definitions [3, 4], hindering rapid scientific advancement [5].

With the growing availability of large-scale image repositories in biology containing millions of images of organisms [6, 7, 8], there is an opportunity for machine learning (ML) methods to discover evolutionary traits automatically from images [5, 9]. This is especially true in light of recent advances in the field of explainable ML, such as the seminal work of ProtoPNet [10] and its variants [11, 12, 13] which find representative patches in training images (termed *prototypes*) capturing discriminatory features for every class. We can thus cast the problem of discovering evolutionary traits into asking the following question: *what image features or prototypes are common across a group of species with a shared ancestor in the tree of life that are absent in species with a different shared ancestor?*

For example, in Figure 1, we can see that the four species of birds on the left descending from the blue node show the common feature of having "long tails," unlike any of the descendant species of the red node. Learning such common features at every internal node as a hierarchy of prototypes can help biologists generate novel hypotheses of species diversification (e.g., the splitting of blue and red nodes) and accumulation of evolutionary trait changes.

Despite the success of ProtoPNet [10] and its variants in learning prototypes over a flat structure of classes, applying them to discover a hierarchy of prototypes is challenging for three main reasons. *First*, existing methods that learn multiple prototypes for every class are prone to learning "over-specific" prototypes at internal nodes of a tree, which cover only one (or a few) of its descendant species. Figure 2 shows a few examples to illustrate the concept of over-specific prototypes. Consider the problem of learning prototypes common to descendant species of the Felidae family: Lion and Bobcat. If we learn one prototype focusing on the feature of the mane (specific only to Lion) and another prototype focusing on the feature of spotted back (specific only to Bobcat), then these two prototypes taken together can classify all images from the Felidae family. However, they do not represent *common* features shared between Lion and Bobcat and hence are not useful for discovering evolutionary traits. Such over-specific prototypes should be instead pushed down to be learned at lower levels of the tree (e.g., the species leaf nodes of Lion and Bobcat).

*Second*, while existing methods such as ProtoPShare [11], ProtoPool [12], and ProtoTree [13] allow prototypes to be shared across classes for re-usability and sparsity, in the problem of discovering evolutionary traits, we want to learn prototypes at an internal node $n$ that are not just shared across all it descendant species but are also absent in the *contrasting set* of species (i.e., species descending from sibling nodes of $n$ representing alternate paths of diversification). *Third*, at higher levels of the tree, finding features that are common across a large number of diverse species is challenging [14, 15]. In such cases, we should be able to abstain from finding common prototypes without hampering accuracy at the leaf nodes—a feature missing in existing methods.

To address these challenges, we present **H**ierarchy aligned **Com**monality through **P**rototypical **Net**works (**HComP-Net**), a framework to learn hierarchical prototypes over the tree of life for discovering evolutionary traits. Here are the main contributions of our work:

1. HComP-Net learns common traits shared by all descendant species of an internal node and avoids the learning of over-specific prototypes in contrast to baseline methods using a novel *overspecificity loss*.

2. HComP-Net uses a novel *discriminative loss* to ensure that the prototypes learned at an internal node are absent in the contrasting set of species with different ancestry.

3. HComP-Net includes a novel *masking module* to allow for the exclusion of over-specific prototypes at higher levels of the tree without hampering classification performance.

4. We empirically show that HComP-Net learns prototypes that are accurate, semantically consistent, and generalizable to unseen species compared to baselines on data from 190 species of birds (CUB-200-2011 dataset) [8], 38 species of fishes [9], and 30 species of butterflies [16]. We show the ability of HComP-Net to generate novel hypotheses about evolutionary traits at different levels of the phylogenetic tree of organisms.

## 2  Related Works

One of the seminal lines of work in the field of prototype-based interpretability methods is the framework of ProtoPNet [10] that learns a set of "prototypical patches" from training images of every class to enable case-based reasoning. Following this work, several variants have been developed such as ProtoPShare [11], ProtoPool [12], ProtoTree [13], and HPnet [17] suiting to different interpretability requirements. Among all these approaches, our work is closely related to HPnet [17], the hierarchical extension of ProtoPNet that learns a prototype layer for every parent node in the tree. Despite sharing a similar motivation as our work, HPnet is not designed to avoid the learning of over-specific prototypes or to abstain from learning common prototypes at higher levels of the tree.

Another related line of work is the framework of PIPNet [18], which uses self-supervised learning methods to reduce the "semantic gap" [19, 20] between the latent space of prototypes and the space of images, such that the prototypes in latent space correspond to the same visual concept in the image space. In HComP-Net, we build upon the idea of self-supervised learning introduced in PIPNet to learn semantically consistent hierarchy of prototypes. Our work is also related to ProtoTree [13], which structures the prototypes as nodes in a decision tree to offer more granular interpretability. However, ProtoTree differs from our work in that it learns the tree-based structure of prototypes automatically from data and cannot handle a known hierarchy. Moreover, the prototypes learned in ProtoTree are purely discriminative and allow for negative reasoning, which is not aligned with our objective of finding common traits of descendant species.

Other related works that focus on finding shared features are ProtoPShare [11] and ProtoPool [12]. Both approaches aim to find common features among classes, but their primary goal is to reduce the prototype count by exploiting similarities among classes, leading to a sparser network. This is different from our goal of finding a hierarchy of prototypes to find evolutionary traits common to a group of species (that are absent from other species).

Outside the realm of prototype-based methods, the framework of Phylogeny-guided Neural Networks (PhyloNN) [9] shares a similar motivation as our work to discover evolutionary traits by representing biological images in feature spaces structured by tree-based knowledge (i.e., phylogeny). However, PhyloNN primarily focuses on the tasks of image generation and translation rather than interpretability. Additionally, PhyloNN can only work with discretized trees with fixed number of ancestor levels per leaf node, unlike our work that does not require any discretization of the tree.

## 3  Proposed Methodology

### 3.1  HComP-Net Model Architecture

Given a phylogenetic tree with $N$ internal nodes, the goal of HComP-Net is to jointly learn a set of prototype vectors $\mathbf{P_n}$ for every internal node $n \in \{1, \ldots, N\}$. Our architecture as shown in Figure 3 begins with a CNN that acts as a common feature extractor $f(x; \theta)$ for all nodes, where $\theta$ represents the learnable parameters of $f$. $f$ converts an image $x$ into a latent representation $Z \in \mathbb{R}^{H \times W \times C}$, where each "patch" at location $(h, w)$ is, $\mathbf{z}_{h,w} \in \mathbb{R}^C$. Following the feature extractor, for every node $n$, we initialize a set of $K_n$ prototype vectors $\mathbf{P_n} = \{\mathbf{p_i}\}_{i=1}^{K_n}$, where $\mathbf{p_i} \in \mathbb{R}^C$. Here, the number of

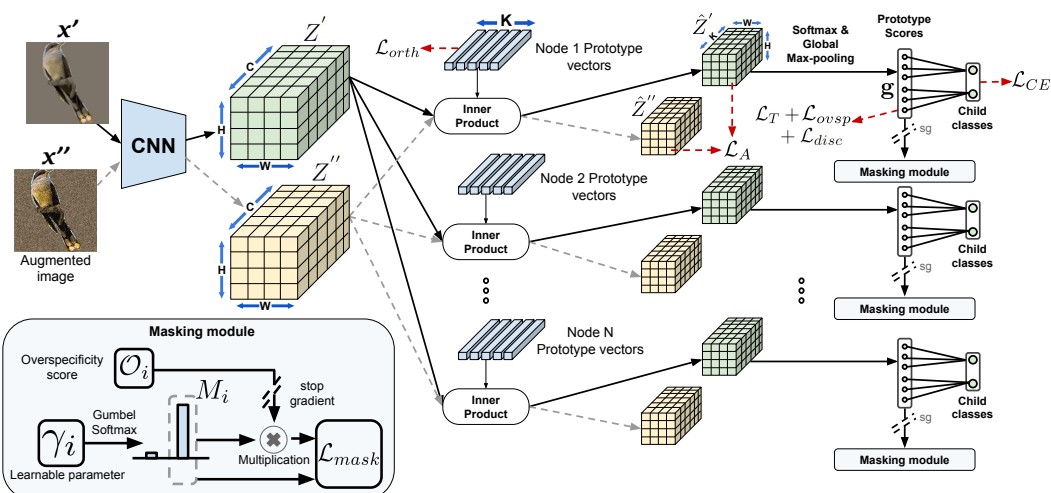

Figure 3: Schematic illustration of HComP-Net model architecture.

prototypes $K_n$ learned at node $n$ varies in proportion to the number of children of node $n$, with $\beta$ as the proportionality constant, i.e., at each node $n$ we assign $\beta$ prototypes for every child node. To simplify notations, we drop the subscript $n$ in $\mathbf{P_n}$ and $K_n$ while discussing the operations occurring in node $n$.

We consider the following sequence of operations at every node $n$. We first compute the similarity score between every prototype in $\mathbf{P}$ and every patch in $Z$. This results in a matrix $\hat{Z} \in \mathbb{R}^{H \times W \times K}$, where every element represents a similarity score between image patches and prototype vectors. We apply a softmax operation across the $K$ channels of $\hat{Z}$ such that the vector $\hat{\mathbf{z}}_{h,w} \in \mathbb{R}^K$ at spatial location $(h, w)$ in $\hat{Z}$ represents the probability that the corresponding patch $\mathbf{z}_{h,w}$ is similar to the $K$ prototypes. Furthermore, the $i^{th}$ channel of $\hat{Z}$ serves as a prototype score map for the prototype vector $\mathbf{p_i}$, indicating the presence of $\mathbf{p_i}$ in the image. We perform global max-pooling across the spatial dimensions $H \times W$ of $\hat{Z}$ to obtain a vector $\mathbf{g} \in \mathbb{R}^K$, where the $i^{th}$ element represents the highest similarity score of the prototype vector $\mathbf{p_i}$ across the entire image. $\mathbf{g}$ is then fed to a linear classification layer with weights $\phi$ to produce the final classification scores for every child node of node $n$. We restrict the connections in the classification layer so that every child node $n_c$ is connected to a distinct set of $\beta$ prototypes, to ensure that every prototype uniquely maps to a child node. $\phi$ is restricted to be non-negative to ensure that the classification is done solely through positive reasoning, similar to the approach used in PIP-Net [18]. We borrow the regularization scheme of PIP-Net to induce sparsity in $\phi$ by computing the logit of child node $n_c$ as $\log((\mathbf{g}\phi)^2 + 1)$. $\mathbf{g}$ and $\phi$ here are again unique to each node.

## 3.2   Loss Functions Used to Train HComP-Net

**Contrastive Losses for Learning Hierarchical Prototypes:** PIP-Net [18] introduced the idea of using self-supervised contrastive learning to learn semantically meaningful prototypes. We build upon this idea in our work to learn semantically meaningful hierarchical prototypes at every node in the tree as follows. For every input image $\mathbf{x}$, we pass in two augmentations of the image, $\mathbf{x}'$ and $\mathbf{x}''$ to our framework. The prototype score maps for the two augmentations, $\hat{Z}'$ and $\hat{Z}''$, are then considered as positive pairs. Since $\hat{\mathbf{z}}_{h,w} \in \mathbb{R}^K$ represents the probabilities of patch $\mathbf{z}_{h,w}$ being similar to the prototypes from $\mathbf{P}$, we align the probabilities from the two augmentations $\hat{\mathbf{z}}'_{h,w}$ and $\hat{\mathbf{z}}''_{h,w}$ to be similar using the following alignment loss:

$$\mathcal{L}_A = -\frac{1}{HW} \sum_{(h,w) \in H \times W} \log(\hat{\mathbf{z}}'_{\mathbf{h,w}} \cdot \hat{\mathbf{z}}''_{\mathbf{h,w}}) \tag{1}$$

Since $\sum_{i=1}^{K} \hat{\mathbf{z}}_{\mathbf{h,w,i}} = 1$ due to softmax operation, $\mathcal{L}_A$ is minimum (i.e., $\mathcal{L}_A = 0$) when both $\hat{\mathbf{z}}'_{\mathbf{h,w}}$ and $\hat{\mathbf{z}}''_{\mathbf{h,w}}$ are identical one-hot encoded vectors. A trivial solution that minimizes $\mathcal{L}_A$ is when all

patches across all images are similar to the same prototype. To avoid such representation collapse, we use the following tanh-loss $\mathcal{L}_T$ of PIP-Net [18], which serves the same purpose as uniformity losses in [21] and [22]:

$$\mathcal{L}_T = -\frac{1}{K} \sum_{i=1}^{K} \log(\tanh(\sum_{b=1}^{B} \mathbf{g_{b,i}})), \tag{2}$$

where $\mathbf{g_{b,i}}$ is the prototype score for prototype $i$ with respect to image $b$ of mini-batch. $\mathcal{L}_T$ encourages each prototype $\mathbf{p_i}$ to be activated at least once in a given mini-batch of $B$ images, thereby helping to avoid the possibility of representation collapse. The use of $\tanh$ ensures that only the presence of a prototype is taken into account and not its frequency.

**Over-specificity Loss:** To achieve the goal of learning prototypes common to all descendant species of an internal node, we introduce a novel loss, termed *over-specificity loss $\mathcal{L}_{ovsp}$* that avoids learning over-specific prototypes at any node $n$. $\mathcal{L}_{ovsp}$ is formulated as a modification of the tanh-loss such that prototype $\mathbf{p_i}$ is encouraged to be activated at least once in every one of the descendant species $d \in \{1, \ldots, D_i\}$ of its corresponding child node in the mini-batch of images fed to the model, as follows:

$$\mathcal{L}_{ovsp} = -\frac{1}{K} \sum_{i=1}^{K} \sum_{d=1}^{D_i} \log(\tanh(\sum_{b \in B_d} \mathbf{g_{b,i}})), \tag{3}$$

where $B_d$ is the subset of images in the mini-batch that belong to species $d$.

**Discriminative loss:** In order to ensure that a learned prototype for a child node $n_c$ is not activated by any of its *contrasting set* of species (i.e., species that are descendants of child nodes of $n$ other than $n_c$), we introduce another novel loss function, $\mathcal{L}_{disc}$, defined as follows:

$$\mathcal{L}_{disc} = \frac{1}{K} \sum_{i=1}^{K} \sum_{d \in \widetilde{D_i}} \max_{b \in B_d}(\mathbf{g_{b,i}}), \tag{4}$$

where $\widetilde{D_i}$ is the contrasting set of all descendant species of child nodes of $n$ other than $n_c$. This is similar to the seperation loss used in other prototype-based methods such as [10], [13], and [23].

**Orthogonality loss:** We also apply kernel orthogonality as introduced in [24] to the prototype vectors at every node $n$, so that the learned prototypes are orthogonal and capture diverse features:

$$\mathcal{L}_{orth} = \|\hat{\mathbf{P}}\hat{\mathbf{P}}^{\top} - I\|_F^2 \tag{5}$$

where $\hat{\mathbf{P}}$ is the matrix of normalized prototype vectors of size $C \times K$, $I$ is an identity matrix, and $\|.\|_F^2$ is the Frobenius norm. Each prototype $\hat{\mathbf{p}}_\mathbf{i}$ in $\hat{\mathbf{P}}$ is normalized as, $\hat{\mathbf{p}}_\mathbf{i} = \frac{\mathbf{p_i}}{\|\mathbf{p_i}\|}$.

**Classification loss:** Finally, we apply cross entropy loss for classification at each internal node as follows:

$$\mathcal{L}_{CE} = -\sum_{b}^{B} y_b \log(\hat{y}_b) \tag{6}$$

where $y$ is ground truth label and $\hat{y}$ is the prediction at every node of the tree.

### 3.3 Masking Module to Identify Over-specific Prototypes

We employ an additional masking module at every node $n$ to identify over-specific prototypes without hampering their training. The learned mask for prototype $\mathbf{p_i}$ simply serves as an indicator of whether $\mathbf{p_i}$ is over-specific or not, enabling our approach to abstain from finding common prototypes if there are none, especially at higher levels of the tree. To obtain the mask values, we first calculate the over-specificity score for prototype $\mathbf{p_i}$ as the product of the maximum prototype scores obtained across all images in the mini-batch belonging to every descendant species $d$ as:

$$\mathcal{O}_i = -\prod_{d=1}^{D_i} \max_{(b \in B_d)}(\mathbf{g_{b,i}}) \tag{7}$$

where $\mathbf{g_{b,i}}$ is the prototype score for prototype $\mathbf{p_i}$ with respect to image $b$ of mini-batch and $B_d$ is the subset of images in the mini-batch that belong to descendant species $d$. Since $\mathbf{g_{b,i}}$ takes a value between 0 to 1 due to the softmax operation, $\mathcal{O}_i$ ranges from -1 to 0, where -1 denotes least

over-specificity and 0 denotes the most over-specificity. The multiplication of the prototype scores ensures that even when the score is less with respect to only one descendant species, the prototype will be assigned a high over-specificity score (close to 0).

As shown in Figure 3, $\mathcal{O}_i$ is then fed into the masking module, which includes a learned mask value $M_i$ for every prototype $\mathbf{p_i}$. We generate $M_i$ from a Gumbel-softmax distribution [25] so that the values are skewed to be very close to either 0 or 1, i.e., $M_i = \text{Gumbel-Softmax}(\gamma_i, \tau)$, where $\gamma_i$ are the learnable parameters of the distribution and $\tau$ is temperature. We then compute the masking loss, $\mathcal{L}_{mask}$, as:

$$\mathcal{L}_{mask} = \sum_{i=1}^{K} (\lambda_{mask} M_i \circ \text{stopgrad}(\mathcal{O}_i) + \lambda_{L_1} \|M_i\|_1) \tag{8}$$

where $\lambda_{mask}$ and $\lambda_{L_1}$ are trade-off coefficients, $\|.\|_1$ is the $L_1$ norm added to induce sparsity in the masks, and $\text{stopgrad}$ represents the stop gradient operation applied over $\mathcal{O}_i$ to ensure that the gradient of $\mathcal{L}_{mask}$ does not flow back to the learning of prototype vectors and impact their training. Note that *the learned masks are not used for pruning the prototypes during training*, they are only used during inference to determine which of the learned prototypes are over-specific and likely to not represent evolutionary traits. Therefore, even if all the prototypes are identified as over-specific by the masking module at an internal node, it will not affect the classification performance at that node.

### 3.4 Training HComP-Net

We first pre-train the prototypes at every internal node in a self-supervised learning manner using alignment and tanh-losses as $\mathcal{L}_{SS} = \lambda_A \mathcal{L}_A + \lambda_T \mathcal{L}_T$. We then fine-tune the model using the following combined loss: $(\lambda_{CE}\mathcal{L}_{CE} + \mathcal{L}_{SS} + \lambda_{ovsp}\mathcal{L}_{ovsp} + \lambda_{disc}\mathcal{L}_{disc} + \lambda_{orth}\mathcal{L}_{orth} + \mathcal{L}_{mask})$, where $\lambda$'s are trade-off parameters. Note that the loss is applied over every node in the tree. We show an ablation of key loss terms in our framework in Table 6 in the Supplementary Section.

## 4 Experimental Setup

**Dataset:** In our experiments, we primarily focus on the 190 species of birds (**Bird**) from the CUB-200-2011 [8] dataset for which the phylogenetic relationship [26] is known. The tree is quite large with a total of 184 internal nodes. We removed the background from the images to avoid the possibility of learning prototypes corresponding to background information such as the bird's habitat as we are only interested in the traits corresponding to the body of the organism. We also apply our method on a fish dataset with 38 species (**Fish**) [9] along with its associated phylogeny [9] and 30 subspecies of Heliconius butterflies (**Butterfly**) from the Jiggins Heliconius Collection dataset [16] collected from various sources [1] along with its phylogeny [52, 53]. The qualitative results of Butterfly and Fish datasets are provided in the supplementary materials. The complete details of hyper-parameter settings and training strategy are also provided in the Supplementary Section E.

**Baselines:** We compare HComP-Net to ResNet-50 [54], INTR (Interpretable Transformer) [55] and HPnet [17]. For HPnet, we used the same hyperparameter settings and training strategy as used by ProtoPNet for CUB-200-2011 dataset. For a fair comparison, we also set the number of prototypes for each child in HPnet to be equal to 10 similar to our implementation. We follow the same training strategy as provided by ProtoPNet for CUB-200-2011 dataset.

## 5 Results

### 5.1 Fine-grained Accuracy

Similar to HPnet [17], we calculate the fine-grained accuracy for each leaf node by calculating the path probability over every image. During inference, the final probability for leaf class $Y$ given an image $X$ is calculated as, $P(Y|X) = P(Y^{(1)}, Y^{(2)}, ..., Y^{(L)}|X) = \prod_{l=1}^{L} P(Y^{(l)}|X)$, where $P(Y^{(l)}|X)$ is the probability of assigning image $X$ to a node at level $l$, and $L$ is the depth of the leaf node. Every image is assigned to the leaf class with maximum path probability, which is used to compute the fine-grained accuracy. The comparison of the fine-grained accuracy calculated for

---

[1]Sources: [27, 28, 29, 30, 31, 32, 33, 34, 35, 36, 37, 38, 39, 40, 41, 42, 43, 44, 45, 46, 47, 48, 49, 50, 51]

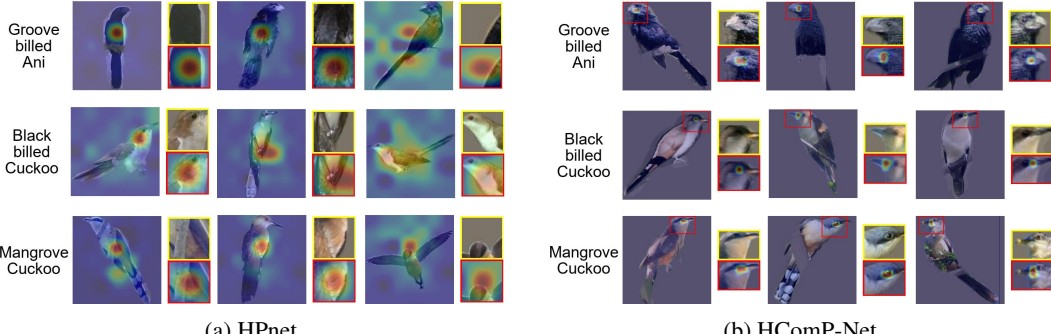

(a) HPnet                                  (b) HComP-Net

Figure 4: Comparing the part consistency of HPnet and HComP-Net for their prototype learned at an internal node in the bird dataset that corresponds to 3 descendant species (names shown on the rows). For every species, we are visualizing the top-3 images with highest prototype score for both HPnet and HComP-Net, shown as the four columns with zoomed in views of their discovered prototypes. We can see that *HPnet highlights varying parts of the bird* across the 3 species and across multiple images of the same species, making it difficult to associate a consistent semantic meaning to its learned prototype. In contrast, *HComP-Net consistently highlights the head region* of the bird across all four species and their images.

HComP-Net and the baselines are given in Table 1. We can see that HComP-Net performs better than the other interpretable methods, such as INTR and HPNet, and is also able to nearly match the performance of non-interpretable models, such as ResNet-50, even outperforming it for the Fish and Butterfly dataset. This shows the ability of our proposed framework to achieve competitive classification accuracy along with serving the goal of discovering evolutionary traits.

Table 1: % Accuracy

| Model | Hierarchy | Bird | Butterfly | Fish |
|---|---|---|---|---|
| ResNet-50 | No | **74.18** | 95.76 | 86.63 |
| INTR |  | 69.22 | 95.53 | 86.73 |
| HPnet | Yes | 36.18 | 94.69 | 77.51 |
| HComP-Net |  | 70.01 | **97.35** | **90.80** |

Table 2: % Accuracy (on unseen species)

| Species Name | HComP-Net | HPnet |
|---|---|---|
| Fish Crow | 53.33 | 10.55 |
| Rock Wren | 53.33 | 10.22 |
| Indigo Bunting | 96.67 | 49.2 |
| Bohemian Waxwing | 70.00 | 44.9 |

## 5.2 Generalizing to Unseen Species in the Phylogeny

We analyze the performance of HComP-Net in generalizing to unseen species that the model hasn't seen during training. The biological motivation for this experiment is to evaluate if HComP-Net can situate newly discovered species at its appropriate position in the phylogeny by identifying its common ancestors shared with the known species. An added advantage of our work is that along with identifying the ancestor of an unseen species, we can also identify the common traits shared by the novel species with known species in the phylogeny. Since unseen species cannot be classified to the finest levels (i.e., up to the leaf node corresponding to the unseen species), we analyze the ability of HComP-Net to classify unseen species accurately up to one level above the leaf level in the hierarchy. With this consideration, the final probability of an unseen species for a given image is calculated as, $P(Y|X_{unseen}) = P(Y^{(1)}, Y^{(2)}, ..., Y^{(L-1)}|X) = \prod_{l=1}^{L-1} P(Y^{(l)}|X)$. Note that we leave out the class probability at the $L^{th}$ level, since we do not take into account the class probability of the leaf level. We leave four species from the Bird training set and calculate their accuracy during inference in Table 2. We can see that HComP-Net is able to generalize better than HPnet for all four species.

## 5.3 Analyzing the Semantic Quality of Prototypes

Following the method introduced in PIPNet [18], we assess the semantic quality of our learned prototypes by evaluating their part purity. A prototype with high part purity (close to 1) is one that consistently highlights the same image region in the score maps (corresponding to consistent local features such as the eye or wing of a bird) across images belonging to the same class. The part

purity is calculated using the part locations of 15 parts that are provided in the CUB dataset. For each prototype, we take the top-10 images from each leaf descendant. We consider the $32 \times 32$ image patch that is centered around the max activation location of the prototype from the top-10 images. With these top-10 image patches, we calculate for each part how frequently the part is present inside

Table 3: Part purity of prototypes on **Bird** dataset.

| Model | $\mathcal{L}_{ovsp}$ | Masking | Part purity | % masked |
|---|---|---|---|---|
| HPnet | - | - | $0.14 \pm 0.09$ | - |
| HComP-Net | - | - | $0.68 \pm 0.22$ | - |
| HComP-Net | - | ✓ | $0.75 \pm 0.17$ | 21.42% |
| HComP-Net | ✓ | - | $0.72 \pm 0.19$ | - |
| HComP-Net | ✓ | ✓ | $\mathbf{0.77 \pm 0.16}$ | 16.53% |

the image patch. For example, a part that is found inside the image patch 8 out of 10 times is given a score of 0.8. In PIP-Net, the highest value among the values calculated for each part is given as the part purity of the prototype. In our approach, since we are dealing with a hierarchy and taking the top-10 from each leaf descendant, a particular part, let's say the eye, might have a score of 0.5 for one leaf descendant and 0.7 for a different leaf descendant. Since we want the prototype to represent the same part for all the leaf descendants, we take the lowest score (the weakest link) among all the leaf descendants as the score of the part. By following this method, for a given prototype we can arrive at a value for each part and finally take the maximum among the values as the purity of the prototype. We take the mean of the part purity across all the prototypes and report the results in Table 3 for different ablations of HComP-Net and also HPnet, which is the only baseline method that can learn hierarchical prototypes.

We can see that HComP-Net, even without the use of over-specificity loss performs much better than HPnet due to the contrastive learning approach we have adopted from PIPNet [18]. The addition of over-specificity loss improves the part purity because over-specific prototypes tend to have poor part purity for some of the leaf descendants which will affect their overall part purity score. Further, for both ablations with and without over-specificity loss, we apply the masking module and remove masked (over-specific) prototypes during the calculation of part purity. We see that the part purity goes higher by applying the masking module, demonstrating its effectiveness in identifying over-specific prototypes. We further compute the purity of masked-out prototypes and notice that the masked-out prototypes have drastically lower part purity ($0.29 \pm 0.17$) compared to non-masked prototypes ($0.77 \pm 0.16$). An alternative approach to learning the masking module is to identify over-specific prototypes using a fixed global threshold over $\mathcal{O}_i$. We show in Table 9 of Supplementary Section F, that given the right choice of such a threshold, we can identify over-specific prototypes. However, selecting the ideal threshold can be non-trivial. On the other hand, our masking module learns the appropriate threshold dynamically as part of the training process.

Figure 4 visualizes the part consistency of prototypes discovered by HComP-Net in comparison to HPnet for the bird dataset. We can see that HComP-Net is finding a consistent region in the image (corresponding to the head region) across all three descendant species and all images of a species, in contrast to HPnet. Futhermore, thanks to the alignment loss, every patch $\hat{\mathbf{z}}_{h,w}$ is encoded as nearly a one-hot encoding with respect to the $K$ prototypes which causes the prototype score maps to be highly localized. The concise and focused nature of the prototype score maps makes the interpretation much more effective compared to baselines.

## 5.4 Analyzing Evolutionary Traits Discovered by HComP-Net

We now qualitatively analyze some of the hypothesized evolutionary traits discovered in the hierarchy of prototypes learned by HComP-Net. Figure 5 shows the hierarchy of prototypes discovered over a small subtree of the phylogeny from Bird (four species) and Fish (three species) dataset. In the visualization of bird prototypes, we can see that the two Pelican species share a consistent region in the learned Prototype labeled 2, which corresponds to the head region of the birds. We can hypothesize this prototype to be capturing the white colored crown common to the two species. On the other hand, Prototype 1 finds the shared trait of similar beak morphology (e.g., sharpness of beaks) across the two Cormorant species. We can see that HComP-Net avoids the learning of over-specific prototypes at internal nodes, which are pushed down to individual leaf nodes, as shown in visualizations of Prototype 3, 4, 5, and 6. Similarly, in the visualization of the fish prototypes, we can see that Prototype 1 is highlighting a specific fin (dorsal fin) of the *Carassius auratus* and *Notropis hudsonius* species, possibly representing their pigmentation and structure, which is noticeably different compared to the contrasting species of *Alosa chrysochloris*. Note that while HComP-Net identifies the common

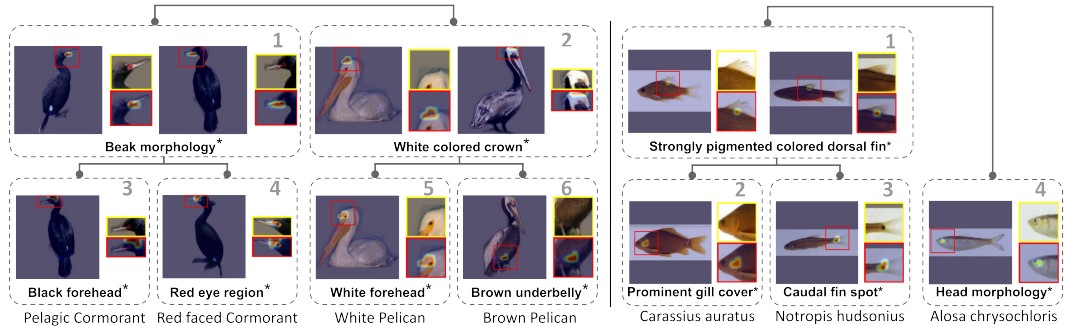

Figure 5: Visualizing the hierarchy of prototypes discovered by HComP-Net for birds and fishes. *Note that the textual descriptions of the hypothesized traits shown for every prototype are based on human interpretation.

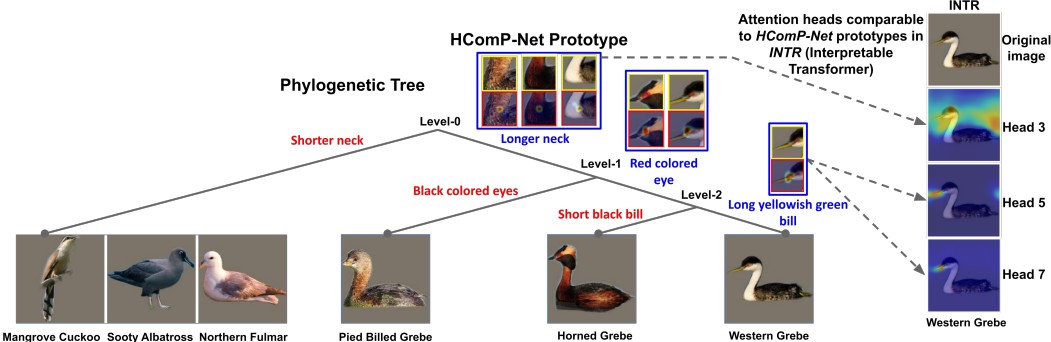

Figure 6: We trace the prototypes learned for Western Grebe at three different levels in the phylogenetic tree (corresponding to different periods of time in evolution). Text in blue is the interpretation of common traits of descendants found by HComP-Net at every ancestor node of Western Grebe.

regions corresponding to each prototype (shown as heatmaps), the textual descriptions of the traits provided in Figure 5 are based on human interpretation.

Figure 6 shows another visualization of the sequence of prototypes learned by HComP-Net for the Western Grebe species at different levels of the phylogeny. We can see that at level 0, we are capturing features closer to the neck region, indicating the likely difference between the length of necks between Grebe species and other species (Cuckoo, Albatross, and Fulmar) that diversify at an earlier time in the process of evolution. At level 1, the prototype is focusing on the eye region, potentially indicating to difference in the color of red and black patterns around the eyes. At level 2, we are differentiating Western Grebe from Horned Grebe based on the feature of bills. We also validate our prototypes by comparing them with the multi-head cross-attention maps learned by INTR [55]. We can see that some of the prototypes discovered by HComP-Net can be mapped to equivalent attention heads of INTR. However, while INTR is designed to produce a flat structure of attention maps, we are able to place these maps on the tree of life. This shows the power of HComP-Net in generating novel hypotheses about how trait changes may have evolved and accumulated across different branches of the phylogeny. Additional visualizations of discovered evolutionary traits for butterfly species and fish species are provided in the supplementary section in Figures 7 to 16.

# 6    Conclusion

We introduce a novel approach for learning hierarchy-aligned prototypes while avoiding the learning of over-specific features at internal nodes of the phylogenetic tree, enabling the discovery of novel evolutionary traits. Our empirical analysis on birds, fishes, and butterflies, demonstrates the efficacy of HComP-Net over baseline methods. Furthermore, HComP-Net demonstrates a unique ability to generate novel hypotheses about evolutionary traits, showcasing its potential in advancing our understanding of evolution. We discuss the limitations of our work in Supplementary Section I. While we focus on the biological problem of discovering evolutionary traits, our work can be applied in general to domains involving a hierarchy of classes, which can be explored in future research.

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

## A  Ablation of Over-specificity Loss Trade-off Hyperparameter

We have provided an ablation for the over-specificity loss trade-off hyperparameter ($\lambda_{ovsp}$) in Table 4. We can observe that increasing the weight of over-specificity loss reduces the model's classification performance, as the model struggles to find any commonality especially at internal nodes where the number of leaf descendant species are large in number and quite diverse. It is natural that species that are diverse and distantly related may share fewer characteristics with each other, in comparison to a set of species that diverged more recently from a common ancestor [14, 15]. Therefore, forcing the model to learn common traits with a strong $\mathcal{L}_{ovsp}$ constraint can cause the model to perform bad in terms of accuracy.

Table 4: Ablation of over-specificity loss trade-off hyperparameter ($\lambda_{ovsp}$). Done on Bird dataset.

| $\lambda_{ovsp}$ | Part purity | Part purity with mask applied | % masked | % Accuracy |
|---|---|---|---|---|
| w/o $\mathcal{L}_{ovsp}$ | $0.68 \pm 0.22$ | $0.75 \pm 0.17$ | 21.42% | 58.32 |
| 0.05 | $\mathbf{0.72 \pm 0.19}$ | $\mathbf{0.77 \pm 0.16}$ | 16.53% | 70.01 |
| 0.1 | $0.71 \pm 0.18$ | $0.74 \pm 0.16$ | 11.31% | **70.97** |
| 0.5 | $0.71 \pm 0.19$ | $0.72 \pm 0.18$ | 4.2% | 68.23 |
| 1.0 | $0.70 \pm 0.19$ | $0.70 \pm 0.2$ | 2.13% | 62.68 |
| 2.0 | $0.69 \pm 0.19$ | $0.69 \pm 0.19$ | 0.55% | 53.16 |

## B  Ablation of Number of Prototypes

In Table 5 we vary the number of prototypes per child $\beta$ for a node to see the impact on model's performance. We note that while the accuracy increases marginally with increasing the number of prototypes per child ($\beta$) from 10 to 15, it also considerably increases the overall number of prototypes initialized. Therefore we continue to work with $\beta = 10$ for all of our experiments.

Table 5: Ablation of number of prototypes per child for a node ($\beta$). Done on Bird dataset.

| Number of Prototypes ($\beta$) | % Accuracy |
|---|---|
| 10 | 70.01 |
| 15 | **70.92** |
| 20 | 67.93 |

## C  Ablation of Individual Losses

In Table 6, we perform an ablation of the various loss terms used in our methodology. As it can be observed, removal of $\mathcal{L}_{ovsp}$ and $\mathcal{L}_{disc}$ degrades performance in terms of both semantic consistency (part purity) and accuracy. On the other hand, removal of self supervised contrastive loss $\mathcal{L}_{SS}$ improves accuracy but at the cost of drastically decreasing the semantic consistency.

Table 6: Ablation of individual losses. Done on Bird dataset.

| Model | Part purity | Part purity with mask applied | % masked | % Accuracy |
|---|---|---|---|---|
| HComP-Net | $0.72 \pm 0.19$ | $0.77 \pm 0.16$ | 16.53% | 70.01 |
| HComP-Net w/o $\mathcal{L}_{ovsp}$ | $0.68 \pm 0.22$ | $0.75 \pm 0.17$ | 21.42% | 58.32 |
| HComP-Net w/o $\mathcal{L}_{disc}$ | $0.69 \pm 0.19$ | $0.72 \pm 0.17$ | 10.95% | 65.99 |
| HComP-Net w/o $\mathcal{L}_{SS}$ | $0.53 \pm 0.18$ | $0.57 \pm 0.15$ | 8.36% | 81.62 |

## D  Consistency of Classification Performance Over Multiple Runs

We trained the model using five distinct random weight initializations. The results showed that the model's fine-grained accuracy averaged 70.63% with a standard deviation of 0.18%.

# E    Implementation Details

We have included all the source code and dataset along with the comprehensive instructions to reproduce the results, in the supplementary material (.zip file).

**Model hyper-parameters:** We build HComP-Net on top of a ConvNeXt-tiny architecture as the backbone feature extractor. We have modified the stride of the max pooling layers of later stages of the backbone from 2 to 1 similar to PIP-Net such that the backbone produces feature maps of increased height and width, in order to get more fine-grained prototype score maps. We implement and experiment our method on ConvNeXt-tiny backbones with $26 \times 26$ feature maps. The length of prototype vectors $C$ is 768. The weights $\phi$ at every node $n$ of HComP-Net are constrained to be non-negative by the use of ReLU activation function [56]. Further, the prototype activation nodes are connected with non-negative weights only to their respective child classes in $W$ while their weights to other classes are made zero and non-trainable.

**Training details:** All models were trained with images resized and appropriately padded to $224 \times 224$ pixel resolution and augmented using TrivialAugment [57] for contrastive learning. The prototypes are pretrained with self-supervised learning similar to PIP-Net for 10 epochs, following which the model is trained with the entire set of loss functions for 60 epochs. We use a batch size of 256 for Bird dataset and 64 for Butterfly and Fish dataset. The masking module is trained in parallel and its training is continued for 15 additional epochs after the training of rest of the model is completed. The trade-off hyper-parameters for the loss functions are set to be $\lambda_{CE} = 2; \lambda_A = 5; \lambda_T = 2; \lambda_{ovsp} = 0.05; \lambda_{disc} = 0.1; \lambda_{orth} = 0.1; \lambda_{mask} = 2.0; \lambda_{L1} = 0.5$. $\lambda_{CE}$, $\lambda_T$ and $\lambda_A$ were borrowed from PIP-Net [18]. Ablations to arrive at suitable $\lambda_{ovsp}$ is provided in Table 4. $\lambda_{disc}$ and $\lambda_{orth}$ were chosen empirically and found to work well on all three datasets. Experiment on unseen species was done by leaving out certain classes from the datasets, so that they are not considered during training.

**Dataset and Phylogeny Details:** Dataset statistics and phylogeny statistics are provided in Table 8 and Table 7 respectively. Bird dataset is created by choosing 190 species from CUB-200-2011 [2] [8] dataset, which were part of the phylogeny. Background from all images were filtered using the associated segmentation metadata [58]. For Butterfly dataset we considered each subspecies as an individual class and considered only the subspecies of genus Heliconius from the Heliconius Collection (Cambridge Butterfly)[3] [16]. There is substantial variation among subspecies of Heliconius species. Furthermore, we balanced the dataset by filtering out the subspecies which did not have 20 or more images. We also sampled a subset of 100 images from each subspecies that had more than 100 images. For Fish [4] dataset, we followed the exact same preprocessing steps as outlined in PhyloNN [9].

**Compute Resources:** The models for Bird dataset were trained on two NVIDIA A100 GPUs with 80GB of RAM each. Butterfly and Fish models were trained on single A100 GPU. As a rough estimate the execution time for training model on Bird dataset is around 2.5 hours. For Butterfly and Fish datasets, the training completes under 1 hour. We used a single A100 GPU during inference stage for all other analysis.

Table 7: High level statistics of the phylogenies used for different datasets.

| Phylogeny | # Internal nodes | Max-depth | Min-depth |
|-----------|------------------|-----------|-----------|
| Bird      | 184              | 25        | 3         |
| Butterfly | 13               | 5         | 2         |
| Fish      | 20               | 11        | 2         |

---

[2]License: CC BY

[3]Note that this dataset is a compilation of images from 25 Zenodo records by the Butterfly Genetics Group at Cambridge University, licensed under Creative Commons Attribution 4.0 International ([27, 28, 29, 30, 31, 32, 33, 34, 35, 36, 37, 38, 39, 40, 41, 42, 43, 44, 45, 46, 47, 48, 49, 50, 51]).

[4]License: CC BY-NC

Table 8: Dataset statistics (# train and validation images).

| Dataset | # Classes | Train set | Validation set |
|---|---|---|---|
| **Bird** | 190 | 5695 | 5512 |
| **Butterfly** | 30 | 1418 | 358 |
| **Fish** | 38 | 4140 | 1294 |

Table 9: Part purity with post-hoc thresholding approach. Done on **Bird** dataset.

| Threshold | Part purity with mask applied | % masked |
|---|---|---|
| 0.2 | $0.74 \pm 0.28$ | 12.28% |
| 0.3 | $0.75 \pm 0.27$ | 13.47% |
| 0.4 | $0.76 \pm 0.26$ | 14.97% |
| 0.5 | $0.77 \pm 0.15$ | 16.66% |
| 0.6 | $0.77 \pm 0.26$ | 17.43% |

## F  Post-hoc Thresholding to Identify Over-specific Prototypes

An alternative approach to learning masking module is to calculate the over-specificity score for each prototype on the test set after training the model. We calculate the over-specificity scores for the prototypes of a trained model as follows,

$$\mathcal{O}_i = -\prod_{d=1}^{D_i} \frac{1}{\text{top}_k} \sum_{i=1}^{\text{top}_k} (\mathbf{g_i}) \tag{9}$$

For a given prototype, we choose the $\text{top}_k$ images with the highest prototype scores from each leaf descendant. After taking mean of the $\text{top}_k$ prototype score, we multiply the values from each descendant to arrive at the over-specificity score for the particular prototype. Subsequently we choose a threshold to determine which prototypes are over-specific. We provide the results of post-hoc thresholding approach that can also be used to identify overspecific prototypes in Table 9. While we can note that this approach can also be effective, validating the threshold particularly in scenarious where there is no part annotations available (such as part location annotation of CUB-200-2011) can be an ardous task. In such cases directly identifying over-specific prototypes as part of the training through masking module can be the more feasible option.

## G  Additional Visualizations of the Hierarchical Prototypes Discovered by HComP-Net

We provide more visualizations of the hierarchical prototypes discovered by HComP-Net for Butterfly (Figures 7 and 8) and Fish (Figure 9) datasets in this section. For ease of visualization, in each figure we visualize the prototypes learned over a small sub-tree from the phylogeny. The prototypes at the lowest level capture traits that are species-specific whereas the prototypes at internal nodes capture the commonality between its descendant species. For Fish dataset, we have provided textual descriptions purely based on human interpretation for the traits that are captured by prototypes at different levels. For Butterfly dataset, since the prototypes are capturing different wing patterns, assigning textual description for them is not straightforward. Therefore, we refrain from providing any text description for the highlighted regions of the learned prototypes and leave it to the reader's interpretation.

## H  Additional Top-K Visualizations of HComP-Net Prototypes

We provide additional top-K visualizations of the prototypes from Butterfly (Figures 10 to 13) and Fish (Figures 14 to 16) datasets, where every row corresponds to a descendant species and the columns corresponds to the top-K images from the species with the largest prototype activation scores. A requirement of a semantically meaningful prototype is that it should consistently highlight the same part of the organisms in various images, provided that the part is visible. We can see in the

figures that the prototypes learned by HComP-Net consistently highlight the same part across all top-K images of a species, and across all descendant species. We additionally show that HComP-Net can find common traits at internal nodes with varying number of descendant species, including 4 species (Figure 10), 5 species (Figures 11 and 12), and 10 species (Figure 13) of butterflies, and 5 species (Figure 14), 8 species (Figure 15) and 18 species (Figure 16) for fish. We also provide several top-k visualizations of prototypes learned for bird species in Figures 17 to 25. This shows the ability of HComP-Net to discover common prototypes at internal nodes of the phylogenetic tree that consistently highlight the same regions in the descendant species images even when the number of descendants is large.

# I  Limitations of Our Work

A fundamental challenge of every prototype-based interpretability method (including ours) is the difficulty in associating a semantic interpretation to the underlying visual concept of a prototype. While some prototypes can be interpreted easily based on visual inspection of prototype activation maps, other prototypes are harder to interpret and require additional domain expertise of biologists. Also, while we have considered large phylogenies as that of the 190 species from CUB dataset, it may still not be representative of all bird species. This limited scope may cause our method to identify apparent homologous evolutionary traits that could differ with the inclusion of more species into the phylogeny. Therefore, our method can be seen as a system that generates potential hypotheses about evolutionary traits discovered in the form of hierarchical prototypes.

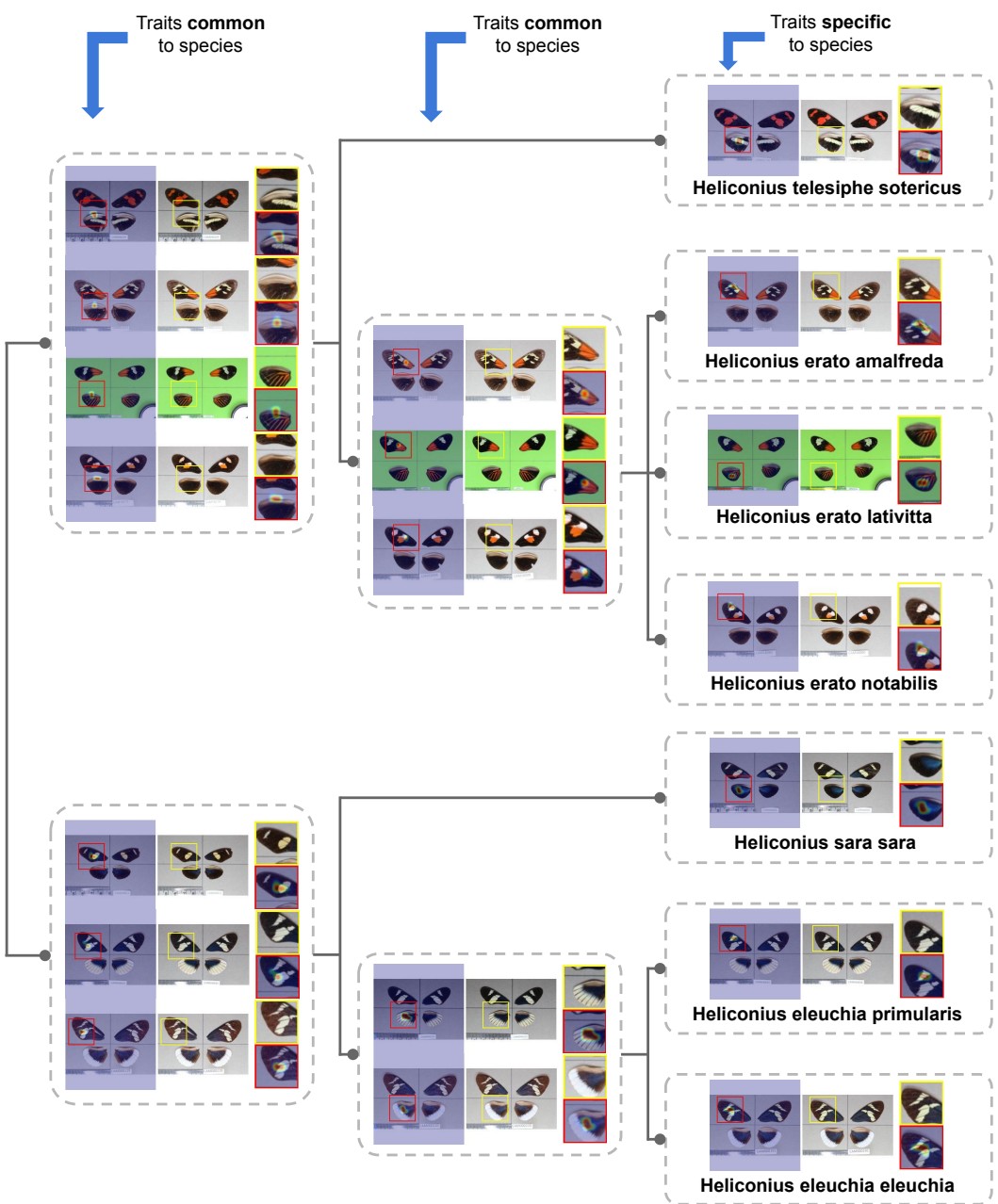

Figure 7: Visualizing the hierarchy of prototypes discovered by HComP-Net over three levels in the phylogeny of seven species from **Butterfly** dataset. For each prototype we visualize one image from each of its leaf descendant. Therefore, for prototypes at species level ( rightmost column) we show only one image whereas for prototypes at internal nodes we show multiple images (equal to the number of leaf descendants). For each image, we show the zoomed in view of the original image as well as the heatmap overlayed image in the region of the learned prototype. The prototypes appear to be capturing different wing patterns of the butterflies.

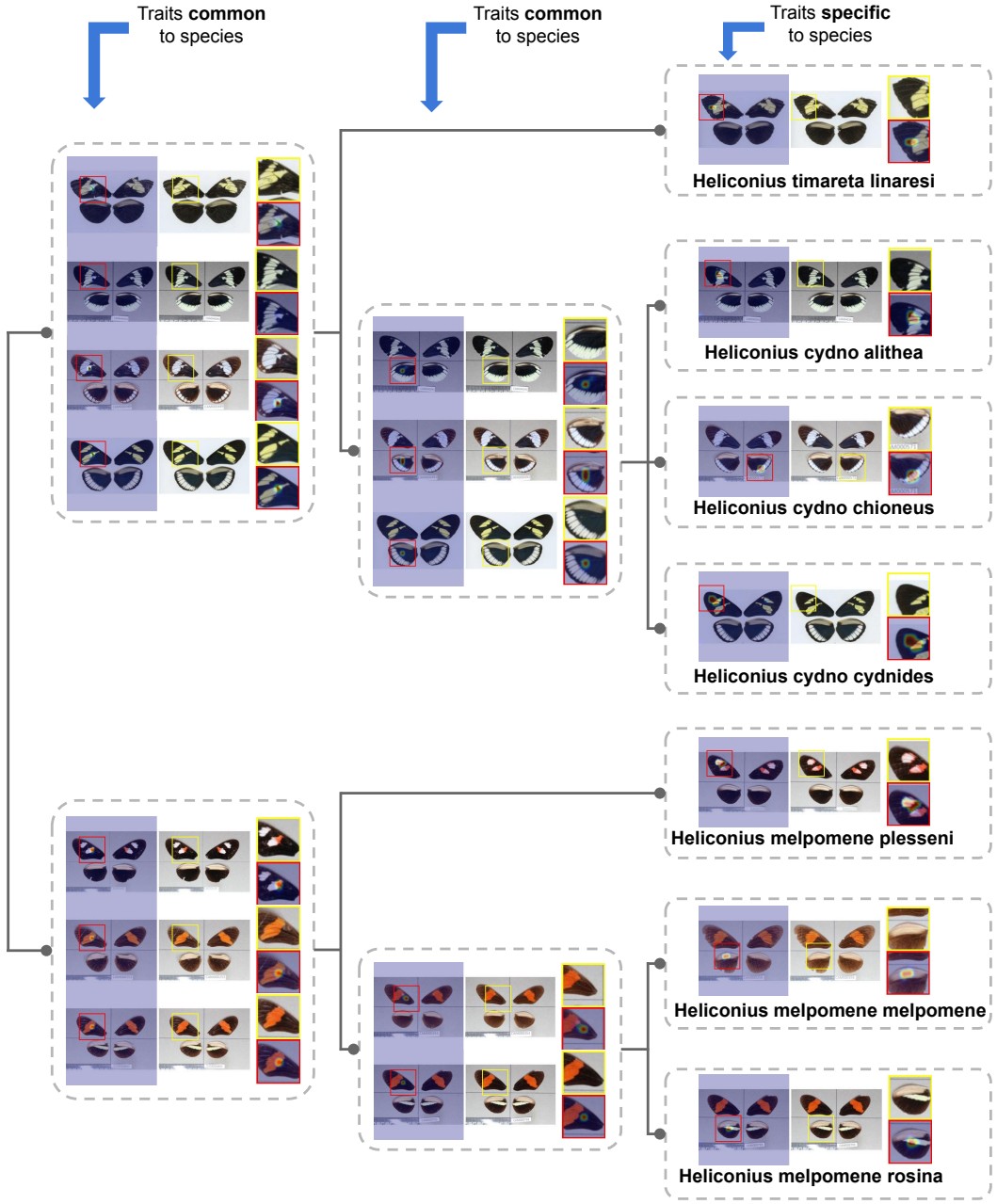

Figure 8: Visualizing the hierarchy of prototypes discovered by HComP-Net over three levels in the phylogeny of seven species from **Butterfly** dataset.

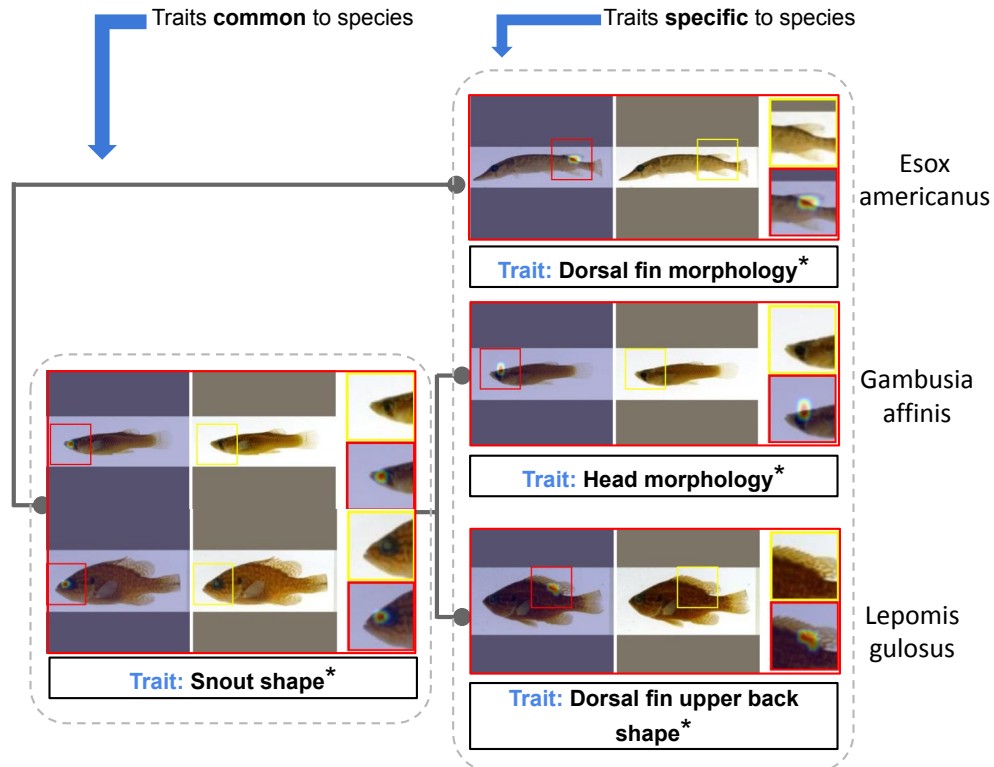

Figure 9: Visualizing the hierarchy of prototypes discovered by HComP-Net for a sub-trees with three species from **Fish** dataset. *Note that the textual descriptions of the hypothesized traits shown for every prototype are based on human interpretation.

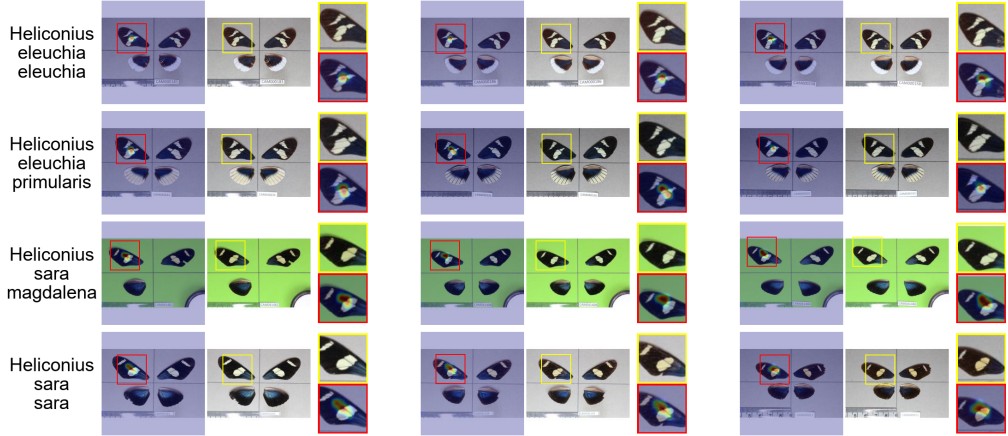

Figure 10: Top-K visualization of a prototype finding commonality between four species of butterfly sharing a common ancestor. Each row represents the top 3 images from the respective species. For each image we show the zoomed in view of the original image as well as the heatmap overlayed image.

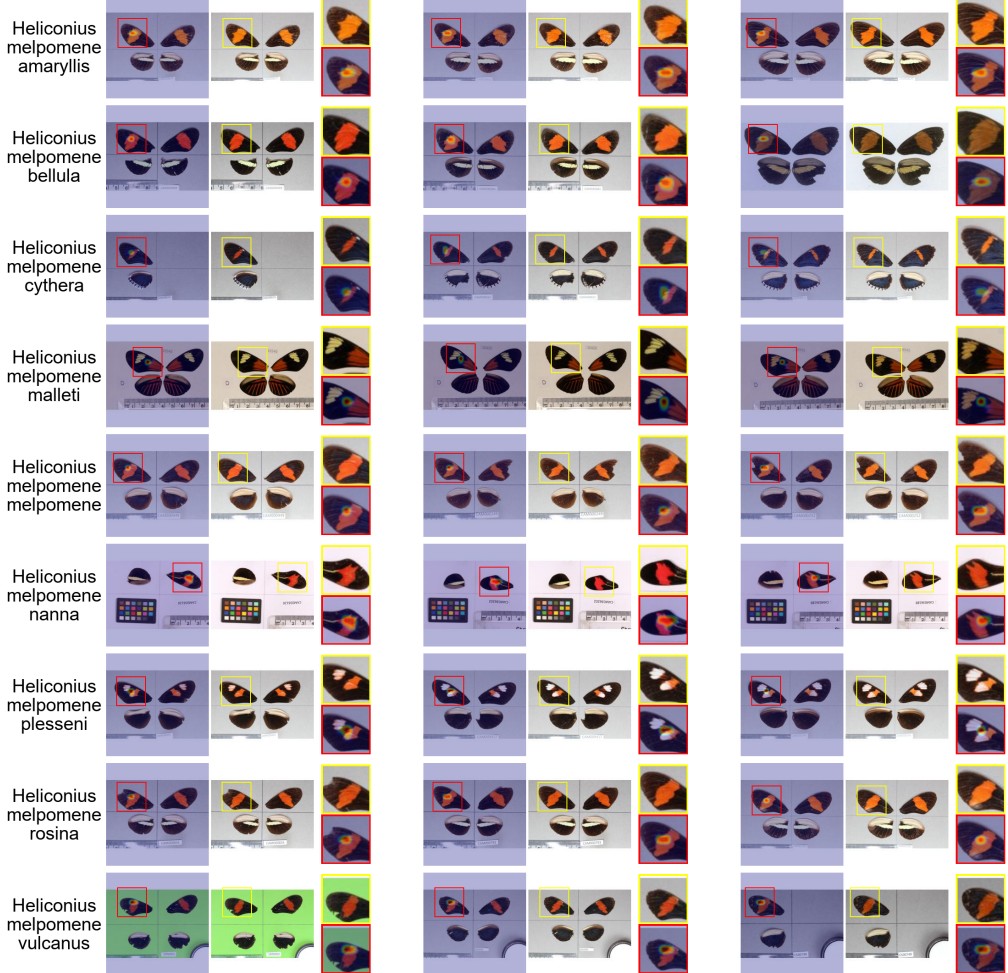

Figure 11: Top-K visualization of a prototype finding commonality between nine species of butterfly sharing a common ancestor. Each row represents the top 3 images from the respective species. For each image we show the zoomed in view of the original image as well as the heatmap overlayed image.

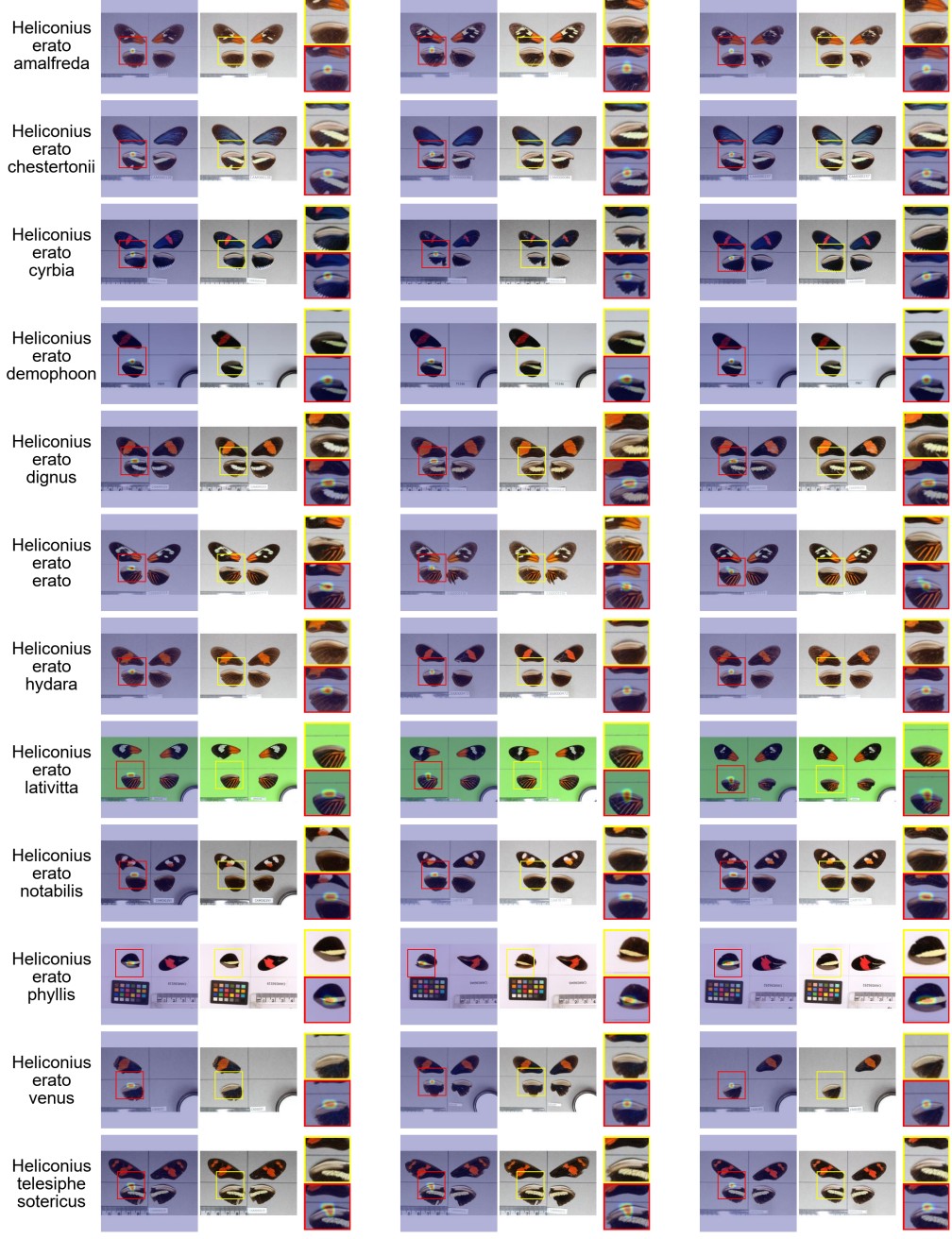

Figure 12: Top-K visualization of a prototype finding commonality between twelve species of butterfly sharing a common ancestor. Each row represents the top 3 images from the respective species. For each image we show the zoomed in view of the original image as well as the heatmap overlayed image.

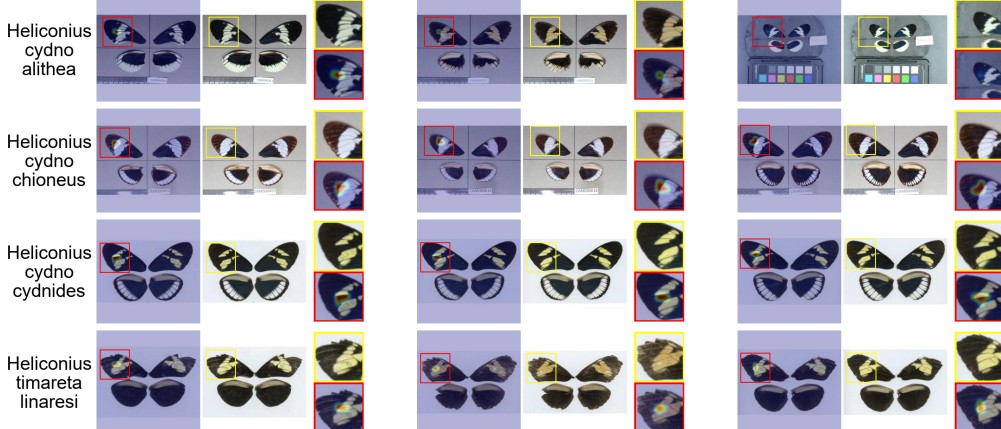

Figure 13: Top-K visualization of a prototype finding commonality between four species of butterfly sharing a common ancestor. Each row represents the top 3 images from the respective species. For each image we show the zoomed in view of the original image as well as the heatmap overlayed image.

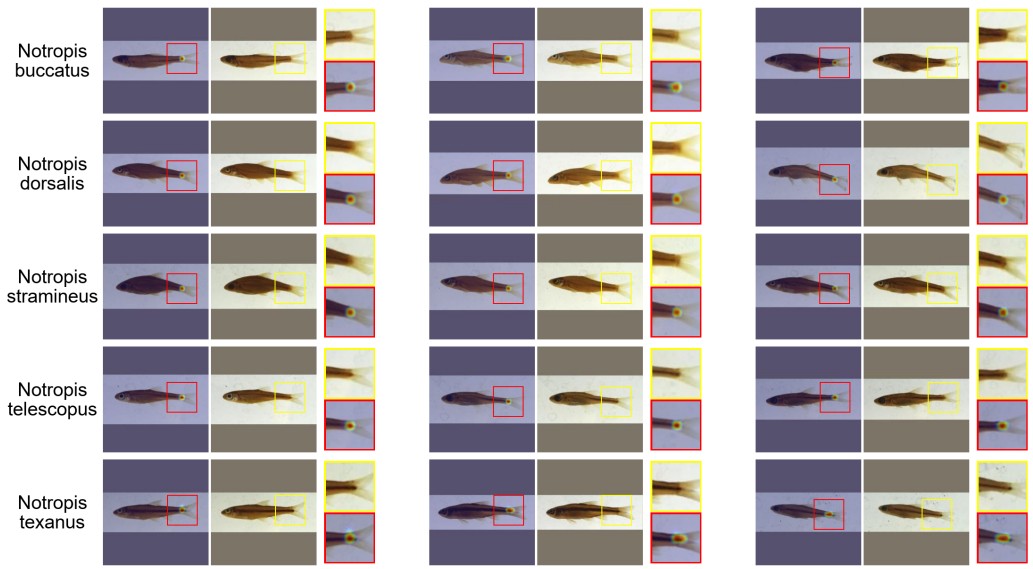

Figure 14: Top-K visualization of a prototype finding commonality between five species of fish sharing a common ancestor. Each row represents the top 3 images from the respective species. For each image we show the zoomed in view of the original image as well as the heatmap overlayed image.

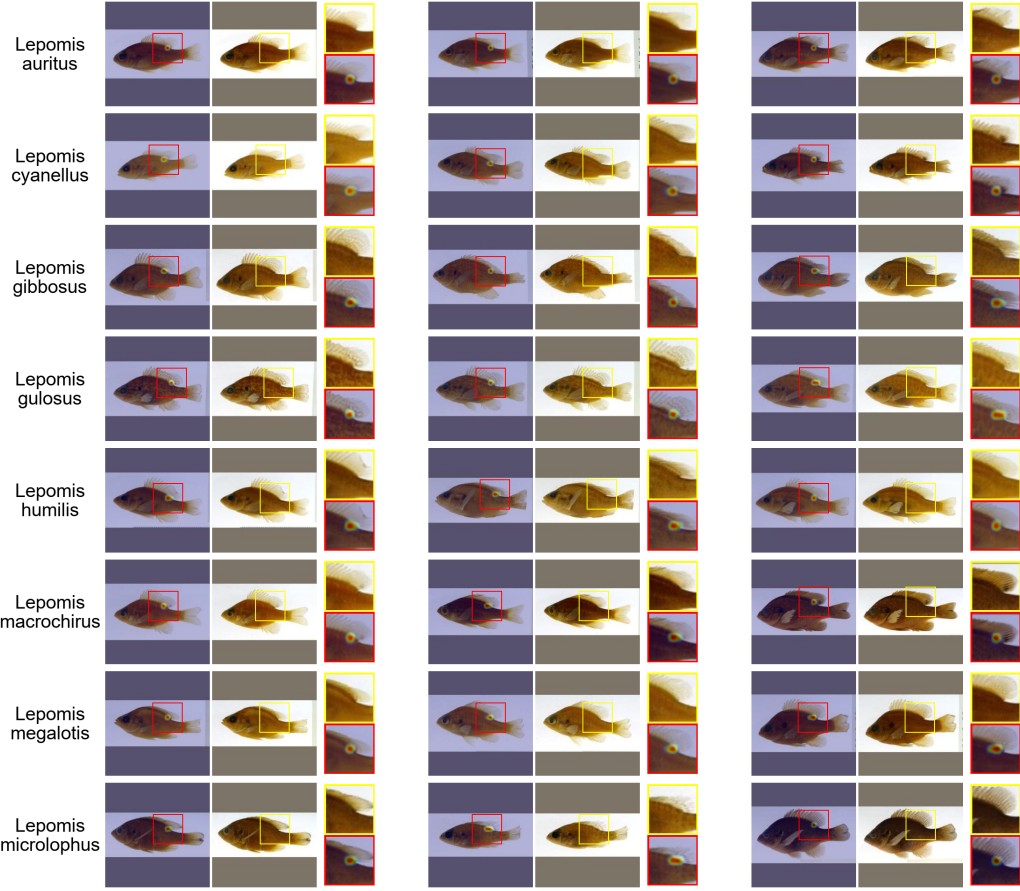

Figure 15: Top-K visualization of a prototype finding commonality between eight species of fish sharing a common ancestor. Each row represents the top 3 images from the respective species. For each image we show the zoomed in view of the original image as well as the heatmap overlayed image.

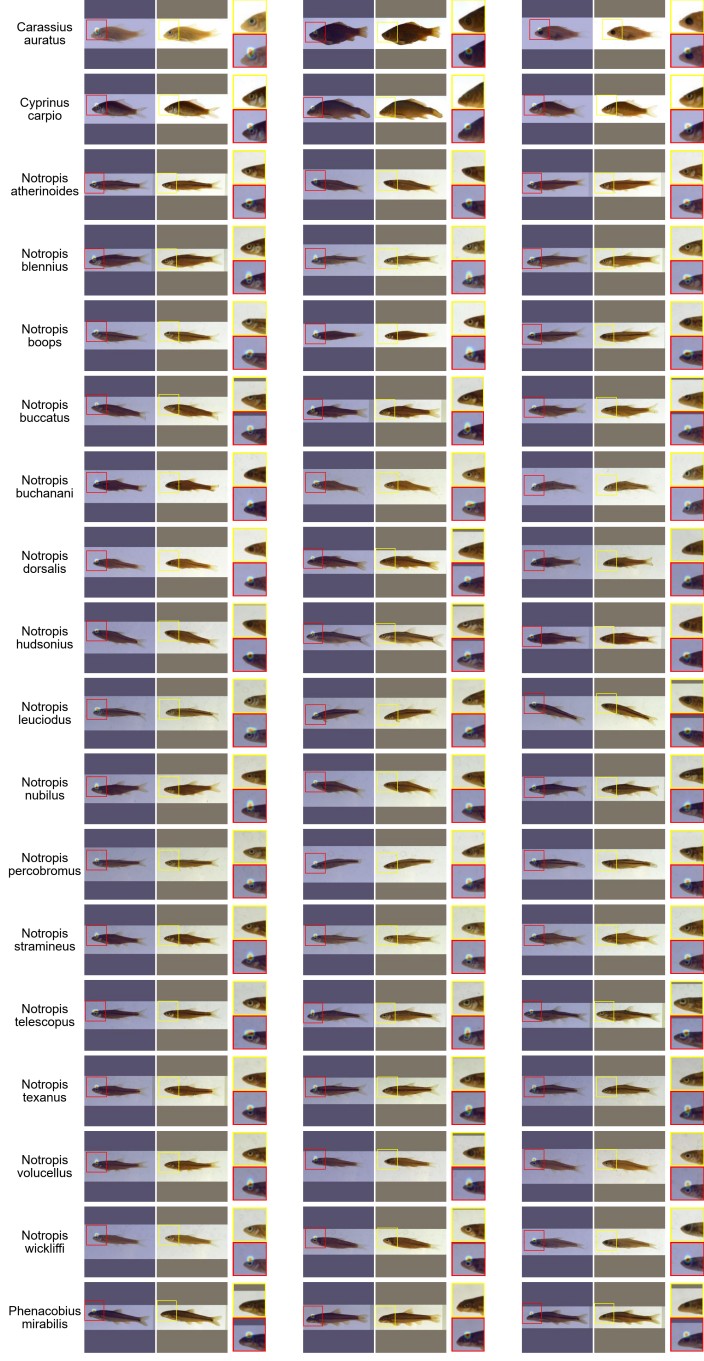

Figure 16: Top-K visualization of a prototype finding commonality between eighteen species of fish sharing a common ancestor. Each row represents the top 3 images from the respective species. For each image we show the zoomed in view of the original image as well as the heatmap overlayed image.

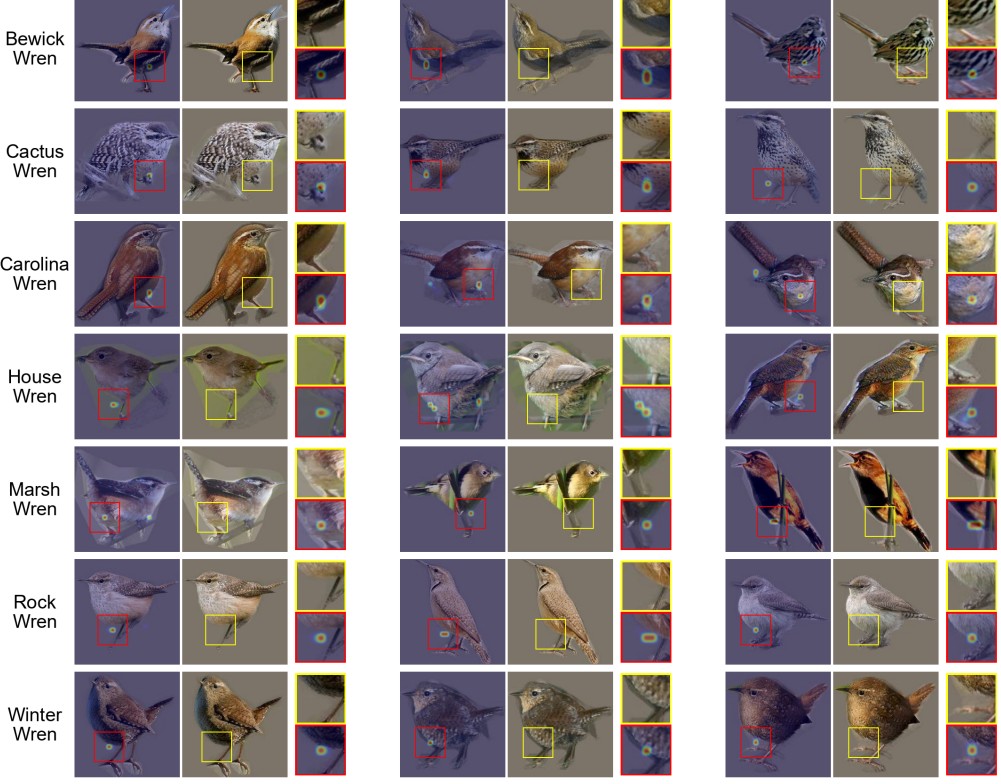

Figure 17: Top-K visualization of a prototype finding commonality between seven species of birds sharing a common ancestor. Each row represents the top 3 images from the respective species. For each image we show the zoomed in view of the original image as well as the heatmap overlayed image.

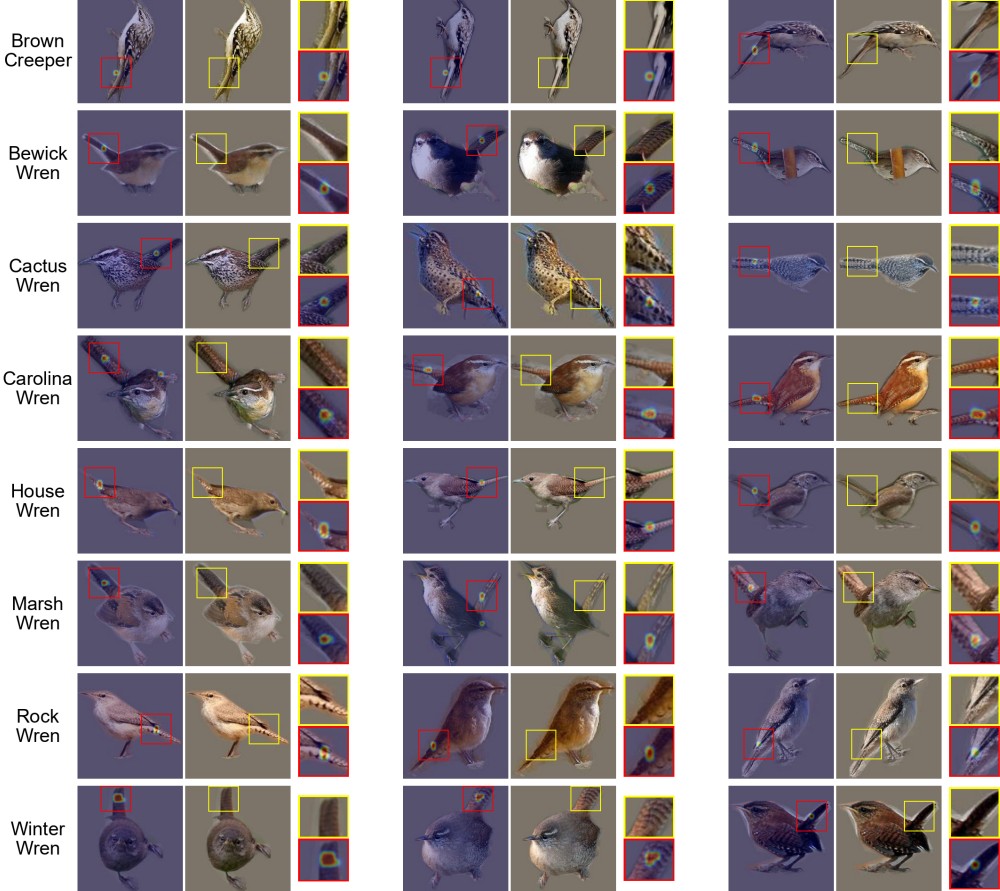

Figure 18: Top-K visualization of a prototype finding commonality between eight species of birds sharing a common ancestor. Each row represents the top 3 images from the respective species. For each image we show the zoomed in view of the original image as well as the heatmap overlayed image.

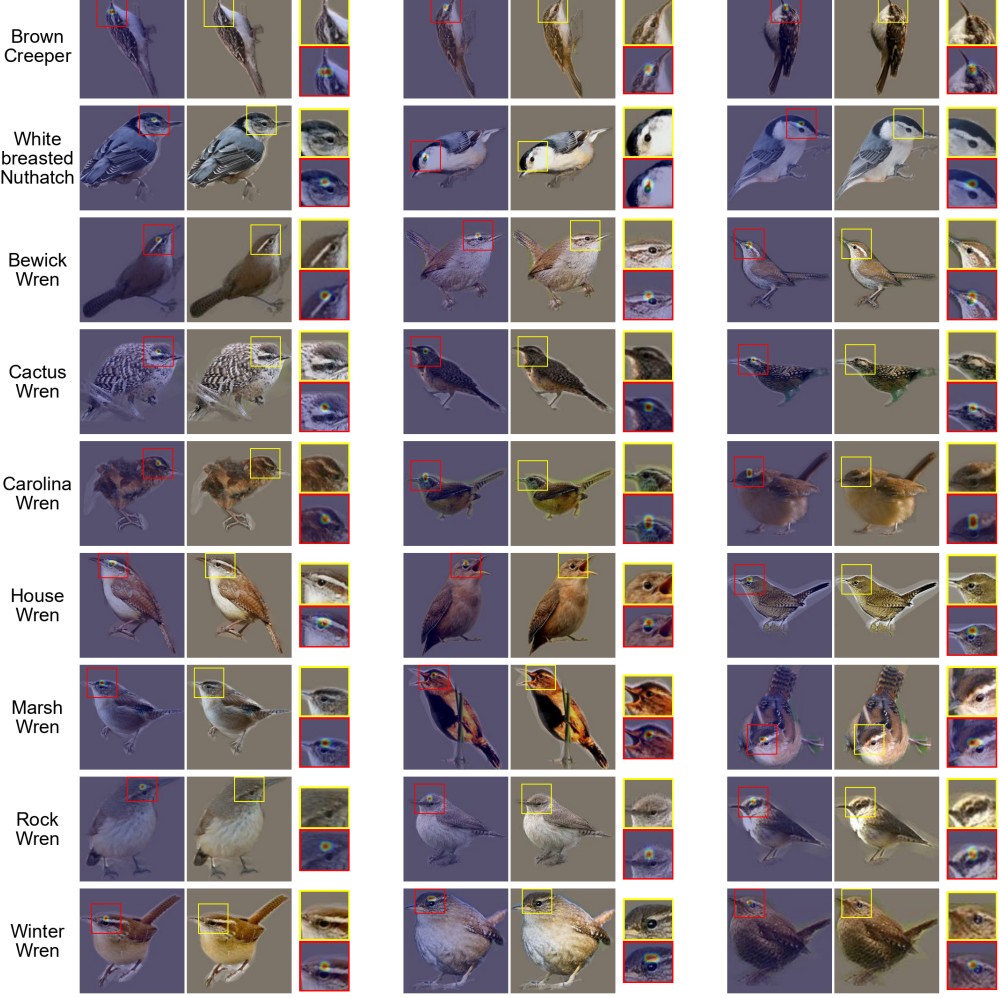

Figure 19: Top-K visualization of a prototype finding commonality between nine species of birds sharing a common ancestor. Each row represents the top 3 images from the respective species. For each image we show the zoomed in view of the original image as well as the heatmap overlayed image.

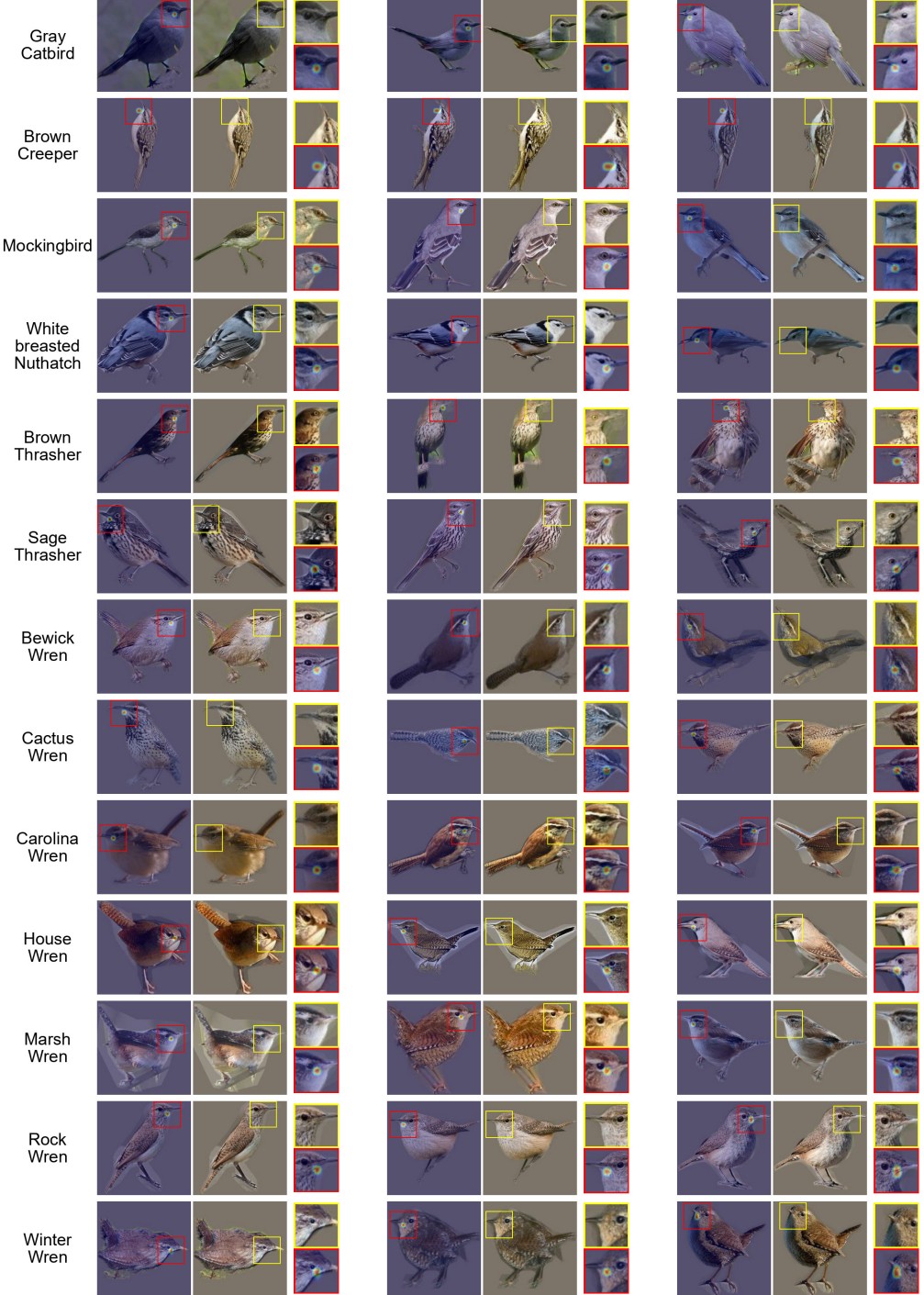

Figure 20: Top-K visualization of a prototype finding commonality between thirteen species of birds sharing a common ancestor. Each row represents the top 3 images from the respective species. For each image we show the zoomed in view of the original image as well as the heatmap overlayed image.

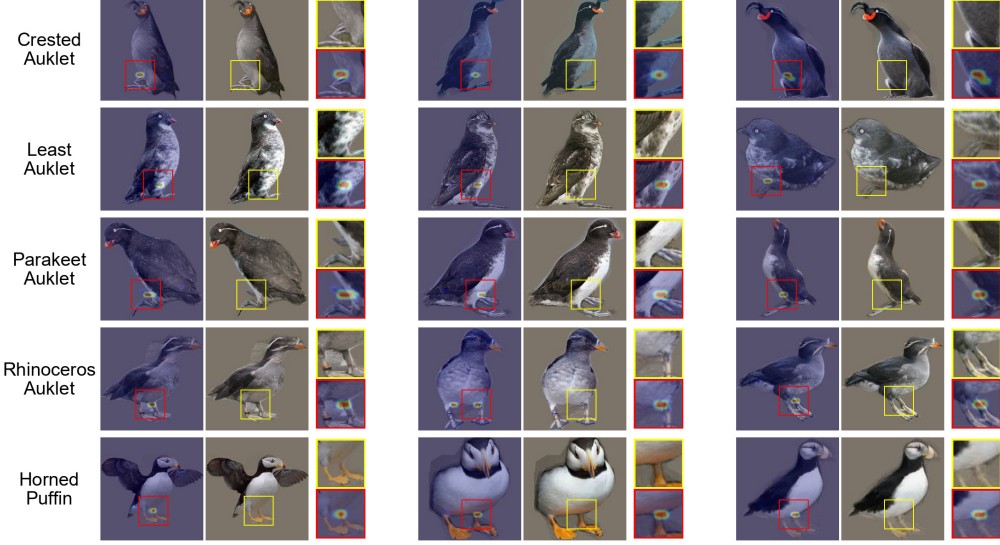

Figure 21: Top-K visualization of a prototype finding commonality between five species of birds sharing a common ancestor. Each row represents the top 3 images from the respective species. For each image we show the zoomed in view of the original image as well as the heatmap overlayed image.

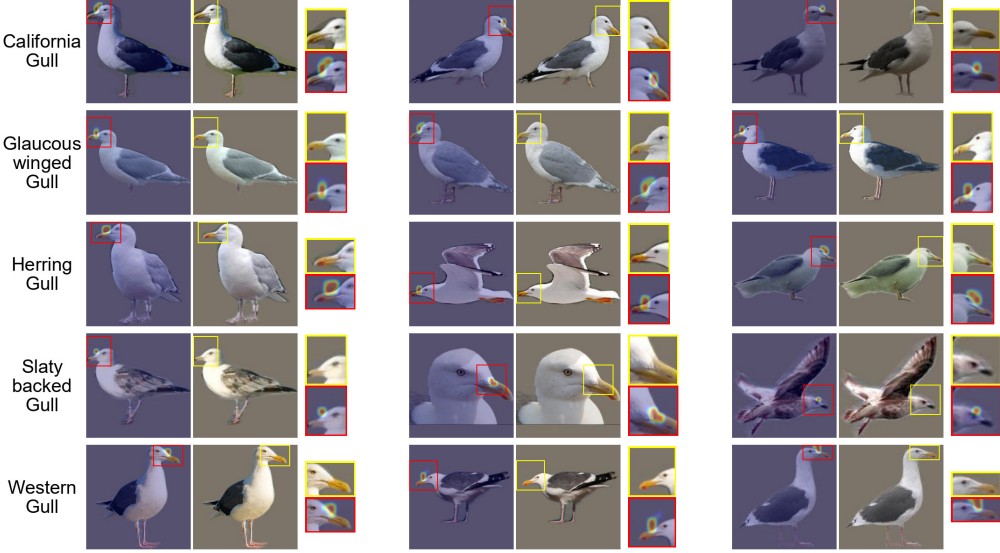

Figure 22: Top-K visualization of a prototype finding commonality between five species of birds sharing a common ancestor. Each row represents the top 3 images from the respective species. For each image we show the zoomed in view of the original image as well as the heatmap overlayed image.

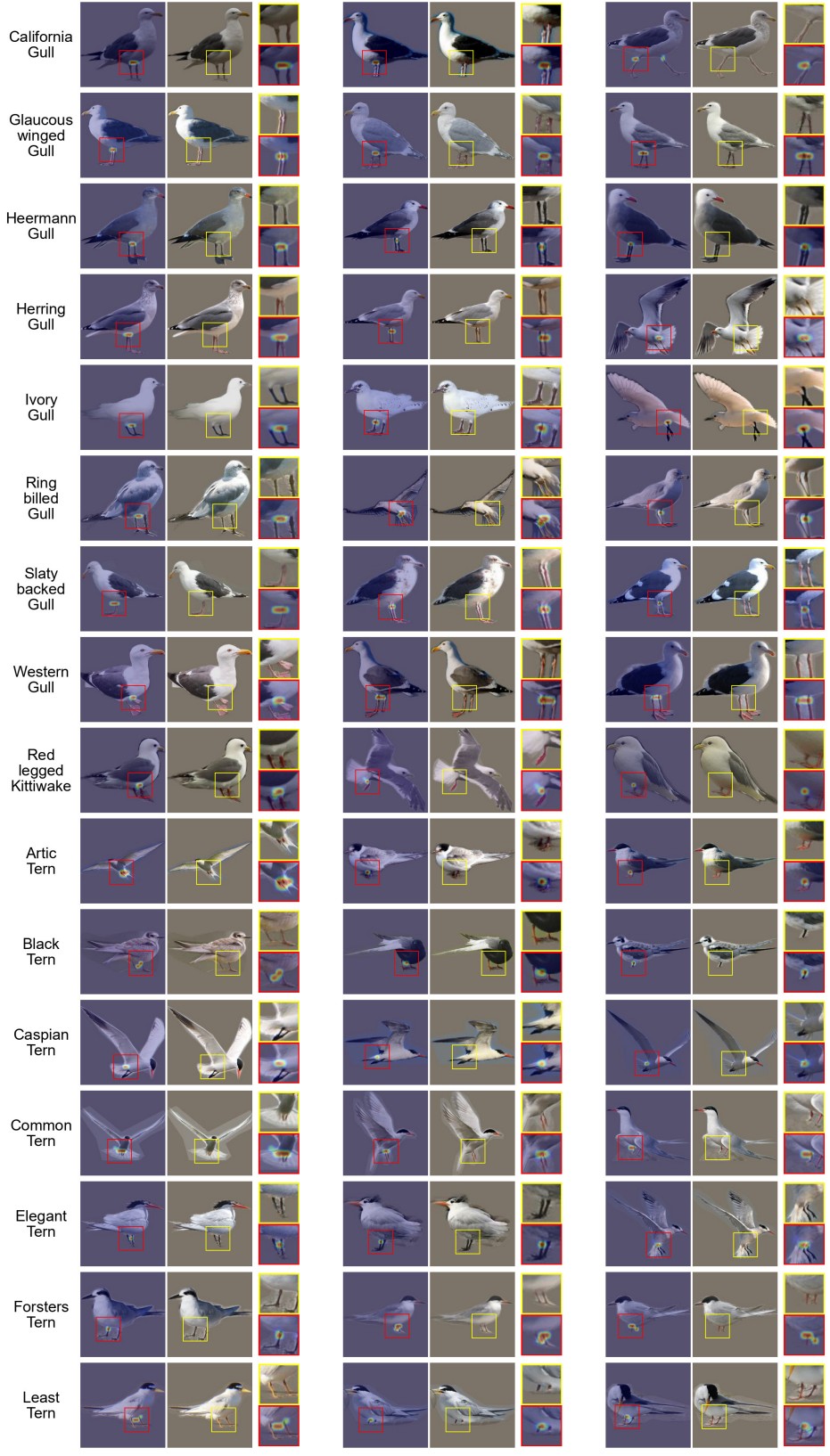

Figure 23: Top-K visualization of a prototype finding commonality between sixteen species of birds sharing a common ancestor. Each row represents the top 3 images from the respective species. For each image we show the zoomed in view of the original image as well as the heatmap overlayed image.

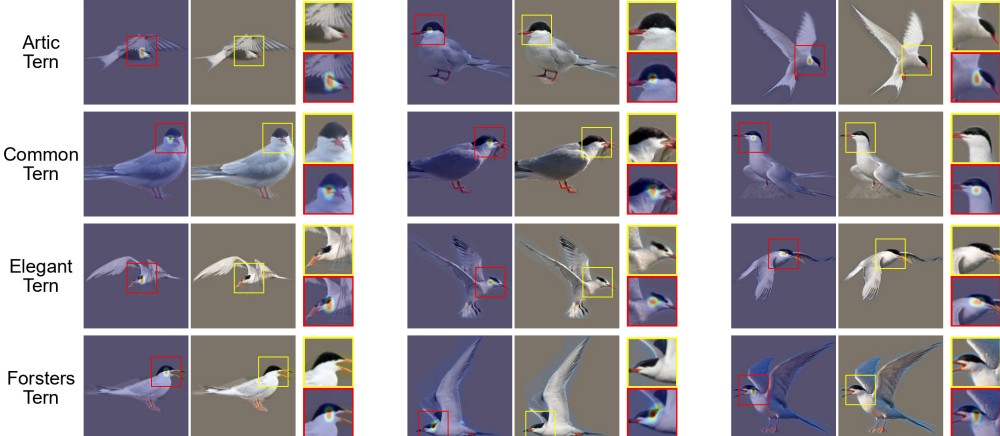

Figure 24: Top-K visualization of a prototype finding commonality between four species of birds sharing a common ancestor. Each row represents the top 3 images from the respective species. For each image we show the zoomed in view of the original image as well as the heatmap overlayed image.

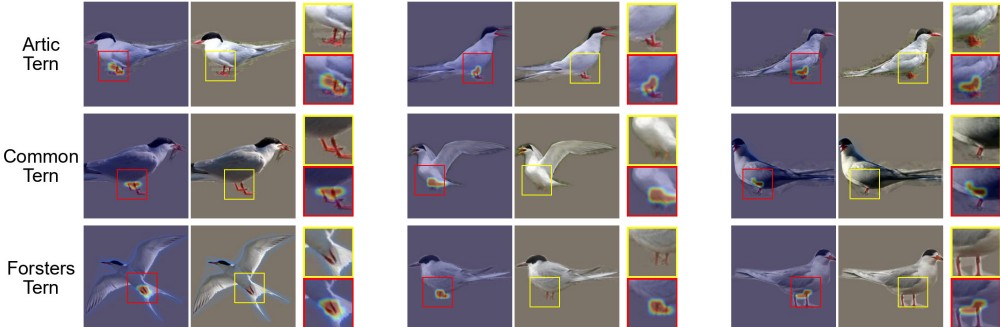

Figure 25: Top-K visualization of a prototype finding commonality between three species of birds sharing a common ancestor. Each row represents the top 3 images from the respective species. For each image we show the zoomed in view of the original image as well as the heatmap overlayed image.

