# OpenReview forum: "What Do You See in Common? Learning Hierarchical Prototypes over Tree-of-Life to Discover Evolutionary Traits"
_NeurIPS.cc/2024/Conference — Submitted to NeurIPS 2024_

### Official Review · Reviewer_hQyP · 2024-07-07

**Soundness:** 2
**Presentation:** 2
**Contribution:** 2
**Rating:** 6
**Confidence:** 2

**Summary:**

The paper addresses a major challenge in biology: identifying evolutionary traits, which are features common to a group of species with a shared ancestor in the phylogenetic tree. Compared to the existing works, this submission proposes new architectures and loss to avoid the over-specification problems. In the experiments, the authors demonstrate that the proposed method improves existing works and set up ablation studies to show the impact of different components of the proposed method.

**Strengths:**

[+] The paper introduces HComP-Net, a new architecture designed to discover evolutionary traits from images in a hierarchical manner. This addresses the limitations of current prototype-based methods that operate over a flat structure of classes.
[+] Together with the architecture, the paper proposes contrastive loss and several additional losses to improve the performance.
[+] The inclusion of a novel masking module allows for the exclusion of over-specific prototypes at higher levels of the tree without compromising classification performance. This helps maintain the accuracy and effectiveness of the model.
[+] The proposed method not only improves the accuracy and other metrics, but also shows the generalizability to unseen species.

**Weaknesses:**

[-] More background: For most of the machine learning conference readers, I guess the proposed problem background is required. Therefore, more related work and background sections should be useful.
[-] I wonder whether the proposed framework can address "Convergent evolution" and other similarity cases. Since these species can have similar features but should not be very close in the evolutionary trees. I suggest the authors to include more details and discussions about the background knowledge.
[-] While the framework has shown promising results on datasets of birds and other animals, I wonder whether the method can show its scalability to larger and more diverse datasets.

**Questions:**

Please refer to the weaknesses.

**Limitations:**

I do not think this work has potential negative social impact. The problems sounds very interesting.

---

> ### Author Rebuttal · Authors · 2024-08-07
>
> We thank the reviewer for the detailed comments and feedback on our work.
>
> **C1**. “More background: For most of the machine learning conference readers, I guess the proposed problem background is required. Therefore, more related work and background sections should be useful.”
>
> > We thank the reviewer for pointing this out. For machine learning readers interested in obtaining a deeper understanding of the biological background, we hope that the following would serve as a good starting point. We will also add this as a dedicated background section in the supplementary.
>
> > One of the first steps in any study of evolutionary morphology is *character construction* - the process of deciding which measurements will be taken of organismal variation that are replicable and meaningful for the underlying biology, and how these traits should be represented numerically [1]. For phylogenetic studies, researchers typically attempt to identify *synapomorphies* – versions of the traits that are shared by two or more species, are inherited from their most recent common ancestor, and may have evolved along the phylogeny branch. The difficulty with the traditional character construction process is that humans often measure traits in a way that is inconsistent and difficult to reproduce, and can neglect shared features that may represent synapomorphies, but defy easy quantification. To address the problem of human inconsistency, PhyloNN [2] and Phylo-Diffusion [3] took a knowledge-guided machine learning (KGML) approach to character construction, by giving their neural networks knowledge about the biological process they were interested in studying (in their case, phylogenetic history), and specifically optimizing their models to find embedded features (analogous to biological traits) that are predictive of that process. To address the problem of visual irreproducibility, Ramirez et. al. [4] suggested photographing the local structures where the empirical traits vary and linking the images to written descriptions of the traits. In this paper, we take influence from both approaches. We extend the hierarchical prototype approach from Hase et. al. [5] to better reflect phylogeny, similar in theory to the way PhyloNN [2] and PhyloDiffusion [3] learned embeddings that reflect phylogeny. Using prototypes, however, we enforce local visual interpretability similar to how researchers may use “type-specimens” to define prototypical definitions of particular character states.
>
>
> **C2**. “I wonder whether the proposed framework can address "Convergent evolution" and other similarity cases. Since these species can have similar features but should not be very close in the evolutionary trees. I suggest the authors to include more details and discussions about the background knowledge.”
>
> > We thank the reviewer for the suggestion. We agree that more discussions about the background biology can indeed be helpful to understand the particular focus of our work. Hereby we give a more detailed description of our work's focus which we will add as part of the introduction in future versions.
>
> > Our method is specifically about finding synapomorphies–shared derived features unique to a particular group of species that share a common ancestor in the phylogeny (referred to as clade). Such features may bear similarities to convergent phenotypes in other clades. However, our goal is not to identify features that exhibit convergence. It is typical for phylogenetic studies to specifically avoid features that exhibit high levels of convergence, as they can lend support for erroneous phylogenetic relationships. Functional studies of trait evolution, however, often target traits that show repeated instances of convergence. While such studies use phylogeny to identify convergent trait evolution, they require additional information to definitively identify convergence. Typically this comes in the form of shared habitat, niche, diet, or behavior. While future iterations of HComP-Net may incorporate such additional information along with phylogeny to identify convergent trait evolution, currently this is beyond the scope of our work.
>
> **C3.** “While the framework has shown promising results on datasets of birds and other animals, I wonder whether the method can show its scalability to larger and more diverse datasets.”
>
> > We kindly request the reviewer to refer to the global comment.
>
> [1] Mezey, J.G. and Houle, D., 2005. The dimensionality of genetic variation for wing shape in Drosophila melanogaster. Evolution, 59(5), pp.1027-1038.
>
> [2] Elhamod, M., Khurana, M., Manogaran, H.B., Uyeda, J.C., Balk, M.A., Dahdul, W., Bakis, Y., Bart Jr, H.L., Mabee, P.M., Lapp, H. and Balhoff, J.P., 2023, August. Discovering Novel Biological Traits From Images Using Phylogeny-Guided Neural Networks. In Proceedings of the 29th ACM SIGKDD Conference on Knowledge Discovery and Data Mining (pp. 3966-3978).
>
> [3] Khurana, M., Daw, A., Maruf, M., Uyeda, J.C., Dahdul, W., Charpentier, C., Bakış, Y., Bart Jr, H.L., Mabee, P.M., Lapp, H. and Balhoff, J.P., 2024. Hierarchical Conditioning of Diffusion Models Using Tree-of-Life for Studying Species Evolution. Accepted for publication at the 18th European Conference on Computer Vision ECCV 2024. arXiv preprint arXiv:2408.00160.
>
> [4] RAMirez, M.J., Coddington, J.A., Maddison, W.P., Midford, P.E., Prendini, L., Miller, J., Griswold, C.E., Hormiga, G., Sierwald, P., Scharff, N. and Benjamin, S.P., 2007. Linking of digital images to phylogenetic data matrices using a morphological ontology. Systematic Biology, 56(2), pp.283-294.
>
> [5] Hase, P., Chen, C., Li, O. and Rudin, C., 2019, October. Interpretable image recognition with hierarchical prototypes. In Proceedings of the AAAI Conference on Human Computation and Crowdsourcing (Vol. 7, pp. 32-40).

---

> > ### Comment · Reviewer_hQyP · 2024-08-13
> > **Thanks for your feedback**
> >
> > These new materials help a lot for the background and experiment setting. I will change my score to weak accept.

---

### Official Review · Reviewer_2ruh · 2024-07-09

**Soundness:** 3
**Presentation:** 3
**Contribution:** 3
**Rating:** 7
**Confidence:** 3

**Summary:**

The authors propose a novel deep learning based algorithm named HComP-Net that can detect evolutionary traits common to groups of species with shared ancestors. Based on earlier studies, they aim to build a model that can accurately isolate common traits of specific species and reject over-specific features.

**Strengths:**

The authors presented their aims and methods quite clearly. While inspired by the earlier studies, they point out how their study is different from the earlier studies. To identify common visual features (i.e., evolutionary traits), they 1) combined two novel loss functions with a previously proposed loss and 2) used a novel masking module. Their results are compelling, which suggest the learning power of HComp-Net and its utility in detecting evolutionary traits. As HComp-Net may be used in other domains, this study can be of interest to other researchers.

**Weaknesses:**

HComp-Net was tested with only 3 datasets, which is understandable, as proper datasets may not be readily available. Still, a more thorough evaluation is desirable in the future.

**Questions:**

I understand each child node is connected to a fixed number of prototypes ($\beta$) in the final classifier. What happens if each child node is connected to all available prototypes? Also, in table 5, can the authors explain why the accuracy is down to 67.93% when $\beta$ is increased to 20?

**Limitations:**

The authors provided the limitations in the appendix.

---

> ### Author Rebuttal · Authors · 2024-08-07
>
> We thank the reviewer for the constructive comments and positive feedback on our work.
>
> **C1**. “HComp-Net was tested with only 3 datasets, which is understandable, as proper datasets may not be readily available. Still, a more thorough evaluation is desirable in the future.”
>
> > We kindly request the reviewer to refer to the global comment.
>
> **C2**. “I understand each child node is connected to a fixed number of prototypes (𝛽) in the final classifier. What happens if each child node is connected to all available prototypes?”
>
> > Sparsity is important for interpretability such that a limited number of prototypes are associated with every class. PIPNet [1] starts with a fully connected structure and then sparsifies the connections during training. This can lead to shared prototypes, i.e., a prototype can be activated for multiple classes. In contrast, we require prototypes to be pre-assigned to classes at the time of initialization. As a result, we do not allow for a prototype to be shared by multiple classes, because we are looking for traits that are common to a group, that *are not* present in the other group.
>
> **C3**. “Also, in table 5, can the authors explain why the accuracy is down to 67.93% when 𝛽 is increased to 20?”
>
> > We thank the reviewer for pointing this out. We verified our results and found that for 𝛽=20, we mistakenly reported the accuracy of the final epoch, while for every other experiment we have reported the best epoch accuracy. We have found the best epoch accuracy for 𝛽=20 is 69.99%. In order to further validate the result, we did four additional runs with different random seed values for this particular experiment and found mean accuracy to be 70.49% with a standard deviation of 0.43%. This shows that the change in 𝛽 does not significantly affect the model performance. We sincerely apologize for this oversight and we will update with the corrected value in the final version.
>
>
> [1] Nauta, M., Schlötterer, J., Van Keulen, M. and Seifert, C., 2023. Pip-net: Patch-based intuitive prototypes for interpretable image classification. In Proceedings of the IEEE/CVF Conference on Computer Vision and Pattern Recognition (pp. 2744-2753).

---

> > ### Comment · Reviewer_2ruh · 2024-08-11
> >
> > I appreciate the authors’ clarifications and would like to keep my rating.

---

> > > ### Author Response · Authors · 2024-08-12
> > >
> > > Thank you very much for your feedback and support.

---

### Official Review · Reviewer_yoRo · 2024-07-12

**Soundness:** 3
**Presentation:** 3
**Contribution:** 3
**Rating:** 7
**Confidence:** 4

**Summary:**

The authors propose a method that automatically learns multiple orthogonal embeddings to act as prototypes. This approach helps the discovery of hierarchical similarities by representing data in a structured space.

**Strengths:**

The writing is clear and easy to follow.
The authors conduct experiments that with other methods, and the visualization of prototypes in feature maps, They also perform an ablation study on different parts of the loss functions.

**Weaknesses:**

In the comparison with HPNet, the authors modify HComP-Net by removing the final two max pooling layers, resulting in a more detailed 26x26 feature map. In contrast, HPNet produces only a 7x7 feature map as shown in figure 4(a). Since the architecture and effectiveness of these networks heavily depend on the resolution of feature maps, this discrepancy raises concerns about the fairness of the comparison. To ensure a fair comparison:
  * HPNet should also be adjusted to generate a larger feature map.
  * This adjustment and its impact on performance should also be included in the ablation study section.

In the generalizing to unseen species section, the evaluation method used by the authors could be extended to include comparisons with non-hierarchical methods. This would provide a more comprehensive evaluation of the method's effectiveness across different types of classification challenges.

**Questions:**

as noted in the weaknesses.

**Limitations:**

There are several limitations concerning the comparison with other methods.

---

> ### Author Rebuttal · Authors · 2024-08-07
>
> We thank the reviewer for the detailed comments and feedback on our work.
>
> **C1.** “In the comparison with HPnet, the authors modify HComP-Net by removing the final two max pooling layers, resulting in a more detailed 26x26 feature map. In contrast, HPnet produces only a 7x7 feature map as shown in figure 4(a). Since the architecture and effectiveness of these networks heavily depend on the resolution of feature maps, this discrepancy raises concerns about the fairness of the comparison. To ensure a fair comparison: HPnet should also be adjusted to generate a larger feature map. This adjustment and its impact on performance should also be included in the ablation study section.”
>
> > We chose 7x7 since it has been used by most methods based on ProtoPNet [2] including HPnet [1]. Since our work is motivated by PIPNet [3], we adapted the use of 26x26 feature maps. Based on the reviewer’s recommendation, we conducted an ablation experiment where we increased the feature map of HPnet to 28x28. We observed the accuracy and part purity to have not improved (results provided in Table 1 of rebuttal document). With better hyper-parameter tuning the method might be able to perform better for 28x28 feature maps, which can be explored as part of future work. We have also provided a qualitative comparison between the HPnet and HComP-Net prototypes with higher resolution feature maps in figure 2 of the rebuttal document. We will add the current results along with the visualizations as part of the supplementary section in the next revision of our paper.
>
> **C2**. “In the generalizing to unseen species section, the evaluation method used by the authors could be extended to include comparisons with non-hierarchical methods. This would provide a more comprehensive evaluation of the method's effectiveness across different types of classification challenges.”
>
> > The definition of path probabilities for unseen species detection (line 235) requires the computation of the probability at internal nodes of the hierarchy. Without having to compute the path probability and directly computing the probability at the leaf node level for non-hierarchical methods, is not directly equivalent to our approach. Further, with hierarchical methods for a given image we get a path that traverses the phylogenetic tree from the root towards the leaf, which is not possible with non-hierarchical methods. Thus, we feel that comparison of the non-hierarchical methods for generalization to unseen species with hierarchical methods would not be a fair comparison.
>
> [1] Hase, P., Chen, C., Li, O. and Rudin, C., 2019, October. Interpretable image recognition with hierarchical prototypes. In Proceedings of the AAAI Conference on Human Computation and Crowdsourcing (Vol. 7, pp. 32-40).
>
> [2] Donnelly, J., Barnett, A.J. and Chen, C., 2022. Deformable protopnet: An interpretable image classifier using deformable prototypes. In Proceedings of the IEEE/CVF conference on computer vision and pattern recognition (pp. 10265-10275).
>
> [3] Nauta, M., Schlötterer, J., Van Keulen, M. and Seifert, C., 2023. Pip-net: Patch-based intuitive prototypes for interpretable image classification. In Proceedings of the IEEE/CVF Conference on Computer Vision and Pattern Recognition (pp. 2744-2753).

---

### Official Review · Reviewer_Ghgm · 2024-07-14

**Soundness:** 3
**Presentation:** 3
**Contribution:** 2
**Rating:** 5
**Confidence:** 3

**Summary:**

The authors investigate the use of prototype-based explainability (as in ProtoPNet) for the visual discovery of evolutionary traits in biology image repositories.
In particular, the authors aim to find traits that apply to group of species in a hierarchical fashion, according ot the tree-of-life hierarchy.
The authors identify three challenges with state-of-the-art prototype methods such as learning over-specific prototypes that do not apply to all species in a given group, and prototypes that do not descriminate between the group and other groups of species in the hierarchy. Ther main contribution is the design of a loss and a masking mechanism to mitigate those issues.

**Strengths:**

+ Interesting application and qualitative results in evolutionary biology.
+ Dedicated focus on learning discriminative hierarchy-level features in an interpretable way.
+ The authors provide their source code.

**Weaknesses:**

- The results are limited to three relatively small datasets. Why are there no result on the iNaturalist dataset which is at least 57x larger than CUB-200- 2011?
- It was hard to judge the effectiveness of the approach from the provided figures. The images are quite small.
- It was also hard to assess the effectiveness of masking. The figures did not illustrate how it helps.

Minor: I encountered several language issues. Below are ones I noted:
- hiearchy
- seperation
- Futhermore
- scenarious => scenarios
- overlayed => overlaid
- indicating to difference => to a difference
- hasn’t => has not [avoid abbreviations in a scientific text]

**Questions:**

- Are the heatmaps simply activation maps from HComPNet? Would GradCAM-based methods be suited to generate fine-grained visualizations?

**Limitations:**

A fundamental limitation in the application domain of interpretable biological traits is discussed in section I. Beyond a few ablation studies focusing on the introduced losses, I missed a discussion on the limitations of the approach, in particular the effectiveness of masking.

---

> ### Author Rebuttal · Authors · 2024-08-07
>
> We thank the reviewer for the detailed comments and feedback on our work.
>
> **C1.** “The results are limited to three relatively small datasets. Why are there no result on the iNaturalist dataset which is at least 57x larger than CUB-200- 2011?”
>
> > We kindly request the reviewer to refer to the global comment.
>
> **C2.** “It was hard to judge the effectiveness of the approach from the provided figures. The images are quite small.”
>
> > We thank the reviewer for their comment on improving the readability of our work. Due to space constraints in the main paper, we had to reduce the size of the images. However, we have ensured that they are of high resolution and can be zoomed in digitally for better clarity. Additionally, we have included larger and high resolution images in the supplementary section for easier visualization and assessment of our results.
>
> **C3.** “It was also hard to assess the effectiveness of masking. The figures did not illustrate how it helps.”
>
> > To understand the nature of the prototypes identified as over-specific by the masking module, we have provided additional qualitative comparison of two prototypes from the same internal node, one of which has been identified as overspecific in Figure 1 of the attached rebuttal document. It can be observed that the over-specific prototype identified by the masking module has lower activation in the heat map as well as poor part purity for one of the species.
>
> > We further quantitatively analyze the effectiveness through the measurement of the part purity of masked out prototypes and unmasked prototypes. In Table 3 of the main paper, we do an ablation study to understand the effect of over-specificity loss and masking on the semantic quality of learned prototypes with part purity as the metric. As we can see, while over-specificity loss improves part purity, the application of masks to remove prototypes that are possibly over-specific further improves the part purity, which we consider as an indicator of the effectiveness of masking. We have also provided the part purity of masked out prototypes in Section 5.3 line 272, to show that the masked out prototypes indeed have a considerably lower part purity.
>
> > We will be including this result and the discussion in the supplementary section of the camera-ready version.
>
> **C4(a).** “Are the heatmaps simply activation maps from HComPNet?”
>
> > The heatmaps are the visualization of the individual channels from prototype-score maps (see Figure 3 of the main paper) called $\hat{Z}$. These prototype score maps are of lower resolutions (26x26), hence we interpolate them into the original image size using bicubic interpolation technique (as is the convention in prototype based methods).
>
> **C4(b).** “Would GradCAM-based methods be suited to generate fine-grained visualizations?”
>
> > GradCAMs [1] are a different class of interpretability methods that offer object based interpretation rather than part based interpretation like prototype-based methods. GradCAMs are not relevant for our work because of the following reasons. First, GradCAMs cannot identify different prototypes corresponding to each trait but only identify a unified discriminative region for a species. Second, there are no current GradCAM based approaches that work on a hierarchy.
>
> **C5.** “A fundamental limitation in the application domain of interpretable biological traits is discussed in section I. Beyond a few ablation studies focusing on the introduced losses, I missed a discussion on the limitations of the approach, in particular the effectiveness of masking.”
>
> > Kindly refer to the response to **C3.** We have provided a qualitative comparison of a masked and unmasked prototype in Figure 1 of the rebuttal document. We also quantitatively analyze the effectiveness of masking by means of part purity in Table 3 of the main paper.
>
> **C6.** “I encountered several language issues”
>
> > Thanks to the reviewer for pointing out the language errors. We will correct them and ensure that no such issues are present in the final version.
>
> [1] Selvaraju, R.R., Cogswell, M., Das, A., Vedantam, R., Parikh, D. and Batra, D., 2017. Grad-cam: Visual explanations from deep networks via gradient-based localization. In Proceedings of the IEEE international conference on computer vision (pp. 618-626).

---

### Author Rebuttal · Authors · 2024-08-07

**General Response to Review Comments**

We sincerely thank all the reviewers for providing constructive feedback. We are encouraged that the reviewers found our work:
- Well-written and easy to follow (Reviewers yoRo, 2ruh)
- Novel and interesting (Reviewers hQyP, 2ruh, Ghgm)
- Shows extensive analysis (Reviewers yoRo, hQyP, 2ruh, Ghgm)

Before we provide individual responses to the main reviewer’s comments, we would like to address a recurring comment, which is **extending our method to other larger datasets**

We are currently exploring the application of HComp-Net on larger datasets like iNaturalist (iNat) [1] but given the limited timeframe for the author rebuttal and the necessary computational resources involved for experimenting on a dataset of that size, we were unable to include the results on iNaturalist. However, we would like to point out that the closest method to our work Phylo-NN [2], which also incorporates the phylogenetic tree, focuses on only one dataset. In contrast, we apply our proposed method to three datasets. These three datasets were chosen based on the biological expertise of our team, allowing us to focus on specific groups of organisms for targeted analysis and validation of results.

While applying HComp-Net to a diverse and large-scale datasets could potentially lead to identifying interesting evolutionary traits spanning a wide variety of species, analyzing them for biologically meaningful information (utilizing the phylogenetic tree) requires extensive domain specific analysis and validation. One of the motivations of HComp-Net is to facilitate interpretable work in *systematic biology*, which seeks to understand evolutionary relationships and trait evolution of species groups. Typically, such studies analyze species that share traits but vary in those traits in ways the researcher wishes to align with evolutionary relationships.  The datasets we used are well-suited to this scale. Larger datasets covering a broader scope of species are challenging to systematize visually because they include species whose traits cannot be easily aligned, *resulting in too much heterogeneity for interpretable analyses*. Therefore, we view our approach as primarily useful for relatively closely related species, which exhibit fine-grained, hard-to-quantify hierarchical variation in visually observable traits.

[1] Van Horn, G., Cole, E., Beery, S., Wilber, K., Belongie, S. and Mac Aodha, O., 2021. Benchmarking representation learning for natural world image collections. In Proceedings of the IEEE/CVF conference on computer vision and pattern recognition (pp. 12884-12893).

[2] Elhamod, M., Khurana, M., Manogaran, H.B., Uyeda, J.C., Balk, M.A., Dahdul, W., Bakis, Y., Bart Jr, H.L., Mabee, P.M., Lapp, H. and Balhoff, J.P., 2023, August. Discovering Novel Biological Traits From Images Using Phylogeny-Guided Neural Networks. In Proceedings of the 29th ACM SIGKDD Conference on Knowledge Discovery and Data Mining (pp. 3966-3978).

---

> ### Author Response · Authors · 2024-08-12
> **Closing Remarks and Request for Final Reviewer Comments and Score Updates**
>
> We sincerely appreciate the insightful comments and valuable feedback provided by our reviewers. As we approach the conclusion of the discussion period, we would like to extend an invitation for any further questions or clarifications. If our responses have adequately addressed your concerns, we kindly request that you consider revising your scores accordingly. Thank you for your time and dedication to this review process.

---

### Decision · Program_Chairs · 2024-09-25

**Decision:**

Reject

**Comment:**

This paper proposes a method to discover evolutionary traits based on a hierarchical prototypical network, which is able to learn prototypes that are semantically consistent and generalize to unseen species. Specifically, the framework learns a set of hierarchical prototypes that correspond to visual features that correspond to common evolutionary traits that are shared across multiple descendent species. The proposed method is validated on CUB dataset and is shown to be effective, and obtain interpretable prototypes.

The paper received positive reviews, as the reviewers found the application as well as the proposed framework to learn discriminative yet hierarchical features as interesting, the masking method to be novel, the paper well-written, and the conducted ablation study to be informative. There were concerns on fairness of the comparison over an existing method (HPNet), lack of experimental results on  relevant benchmarks (e.g. iNaturalist).

Although the review scores are positive, in the meta-reviewer's point of view they are mostly uninformative and the paper lacks novelty in both the high-level idea and the specific design of the framework for learning hierarchical prototypes. The idea of using semantic ontologies to identify discriminative features at different semantic granularity has been revisited many times in the last decade in the computer vision community. However, the paper misses out discussions or comparative studies against many relevant works (a few of them listed below) that aim to achieve the same goal of learning hierarchical features.

[Yan et al. 15] HD-CNN: Hierarchical Deep Convolutional Neural Networks for Large Scale Visual Recognition, ICCV 2015
[Hu et al. 16] Learning Structure Inference Neural Networks, CVPR 2016
[Goo et al. 16] Taxonomy-Regularized Semantic Deep Convolutional Networks, ECCV 2016